# Rewiring of the promoter-enhancer interactome and regulatory landscape in glioblastoma orchestrates gene expression underlying neurogliomal synaptic communication

Chaitali Chakraborty[1,2,3], Itzel Nissen [1,2,3], Craig A. Vincent[1,2], Anna-Carin Hägglund[1,2], Andreas Hörnblad [1] & Silvia Remeseiro [1,2] ✉

Chromatin organization controls transcription by modulating 3D-interactions between enhancers and promoters in the nucleus. Alterations in epigenetic states and 3D-chromatin organization result in gene expression changes contributing to cancer. Here, we map the promoter-enhancer interactome and regulatory landscape of glioblastoma, the most aggressive primary brain tumour. Our data reveals profound rewiring of promoter-enhancer interactions, chromatin accessibility and redistribution of histone marks in glioblastoma. This leads to loss of long-range regulatory interactions and overall activation of promoters, which orchestrate changes in the expression of genes associated to glutamatergic synapses, axon guidance, axonogenesis and chromatin remodelling. SMAD3 and PITX1 emerge as major transcription factors controlling genes related to synapse organization and axon guidance. Inhibition of SMAD3 and neuronal activity stimulation cooperate to promote proliferation of glioblastoma cells in co-culture with glutamatergic neurons, and in mice bearing patient-derived xenografts. Our findings provide mechanistic insight into the regulatory networks that mediate neurogliomal synaptic communication.

Gene regulation critically relies on regulatory sequences such as enhancers, which control the spatial and temporal specificity of gene expression. In vertebrates, enhancers are often located hundreds of kilobases (kb) to even megabases (Mb) away from their target gene promoters. This long-range action of enhancers over gene promoters is facilitated via the structural organization of the genome in the 3D nuclear space, mainly within sub-megabase domains known as TADs (topologically associating domains)[1,2], which emerge from multiple nested loops by loop extrusion mechanisms[3]. This topological organization of the chromatin allows for physical proximity between enhancers and promoters, constraining the range of action of enhancers and therefore setting the stage for the specificity of promoter-enhancer interactions. Enhancer activity is intimately linked to epigenetic status and, thus, enhancers display different states (i.e., active, silenced, primed or poised) depending on the combination of histone marks and other chromatin features[4–8]. Genome-wide enhancer maps have linked risk variants to disease genes[9], and most genomic variants that predispose to cancer are located in non-coding regions

[1]Umeå Centre for Molecular Medicine (UCMM), Umeå University, Umeå, Sweden. [2]Wallenberg Centre for Molecular Medicine (WCMM), Umeå University, Umeå, Sweden. [3]These authors contributed equally: Chaitali Chakraborty, Itzel Nissen. ✉e-mail: silvia.remeseiro@umu.se

with potential to act as *cis*-regulatory elements[10]. Various evidence supports the role of alterations in the regulatory and topological landscape in different types of cancer. Aberrant super-enhancer function provides oncogenic properties, and cancer cells can acquire super-enhancers to drive expression of oncogenes[11–13]. Proto-oncogenes can become activated by disruption of chromosome insulated neighbourhoods[14], and systematic occurrence of structural rearrangements in *cis*-regulatory elements (e.g., enhancer hijacking) mediates dysregulation in cancer[15].

Glioblastoma (GB) (WHO grade 4 astrocytoma) is the most malignant and aggressive form in the wide spectrum of gliomas, the most common primary brain tumours. With a 5-year survival rate of only 3–4%[16], GB prognosis has not improved considerably in the last decades. Glioblastomas develop rapidly and manifest after a short clinical history of usually less than 3 months[17]. No uniform aetiology has been identified, and mechanistic understanding of GB initiation and progression is difficult given the complexity of genomic, epigenomic, metabolic and microenvironment events contributing to disease. Extensive inter- and intra-tumour heterogeneity are characteristics of GB that have complicated the finding of specific and targeted therapies.

Epigenetic alterations play a central role in the aetiology of gliomas and are used for molecular classification[18]. In higher-grade gliomas, diverse alterations in genes encoding chromatin remodellers and epigenetics-related enzymes have been described in conjunction with deregulation of the epigenetic landscape and subsequent gene expression alterations[19–21]. Early studies identified four GB subtypes defined by aberrations and gene expression changes in *EGFR*, *NF1*, and *PDGFRA/IDH1*, corresponding to classical, mesenchymal and neural/proneural subtypes, respectively[22]. Single-cell studies have recently described the plasticity of GB cells that can transition between four main cellular states influenced by the tumour microenvironment[23], as well as a gradient of developmental and wound-response cell states in glioblastoma stem cells (GSCs)[24]. In recent years, researchers have begun to integrate transcriptomic analysis with chromatin and epigenetic profiles, DNA methylomes or chromatin architecture and chromatin accessibility data in glioblastomas[25–31], evidencing that GB is a heterogeneous entity distinguishable from lower grade gliomas. However, despite the efforts to further classify GB into molecular subtypes and to identify relevant subpopulations, this has not yet been translated into a clinical benefit.

In this unsupervised study, we apply a multi-omics approach to map the promoter-enhancer interactome and the regulatory landscape of GB, including histone marks and chromatin accessibility, in a panel of 15 patient-derived GB cell lines representing all four expression subtypes, together with normal human astrocytes and oligodendrocyte progenitor cells (OPCs) as controls. We observe a rewiring of the promoter-enhancer interactions, changes in chromatin accessibility and a redistribution of histone marks across all four expression subtypes in GB. These changes in the regulatory and topological landscapes lead to a significant loss of long-range regulatory interactions and an overall activation of promoter-hubs. This orchestrates changes in the expression of genes associated to synapses, in particular glutamatergic synapses, as well as axon guidance and axonogenesis, and chromatin binding/remodelling. Motif search analysis and CUT&RUN reveal the transcription factors (TFs) SMAD3 and PITX1 as major direct regulators of a set of downstream target genes related to synaptic contacts and axon guidance. In addition, we functionally demonstrate that inhibition of SMAD3 and stimulation of neural activity additively cooperate to promote proliferation of GB cells. The findings reported here are primarily non-hypothesis-driven results that fit well with the recent findings in the emerging field of cancer neuroscience. After the recent discovery of neurogliomal synapses[32–34], that drive tumour progression by relaying neuronal activity to tumour cells, our data offers mechanistic insight into the gene regulatory networks that mediate the neurogliomal synaptic communication in glioblastoma.

## Results

### A map of the regulatory landscape and promoter-enhancer interactome in GB

To obtain a map of the enhancer landscape and promoter-enhancer (P-E) interactome in GB, we performed a multi-omics approach in a panel of 15 patient-derived GB cell lines alongside normal human astrocytes and OPCs as controls (Fig. 1a). The 15 patient-derived GB cell lines were obtained through the human glioblastoma cell culture (HGCC) resource[35], a panel of newly established and well characterized glioblastoma lines derived from GB patient surgical samples, that represent all four expression subtypes (i.e., classical, mesenchymal, neural and proneural, Supplementary Fig. 1). Normal human astrocytes and OPCs were selected as controls since they are considered to be cells of origin for GB[36–41].

In this unsupervised study, we mapped the promoter-enhancer interactome by HiChIP[42], a protein-centric chromatin conformation capture method, using an antibody specific to the promoter mark H3K4me3. Assay for transposase-accessible chromatin with sequencing (ATAC-seq)[43] was used to map chromatin accessibility genome-wide. Via ChIP-seq (Chromatin ImmunoPrecipitation-sequencing) we determined enrichment for the histone modifications: H3K27ac, an active enhancer mark; H3K27me3, a repressive mark; and H3K4me3, predominantly associated to gene promoters. Transcriptome profiling of each line was done by RNA-seq. Our data constitutes a comprehensive map of the regulatory landscape in GB across all four subtypes, and provides functionally relevant topological information by mapping the promoter-enhancer interactome (Fig. 1b). In the next sections, we compare the regulatory and topological landscape of GB to the control astrocytes and OPCs, while focusing on the features present in all four GB subtypes.

### Gene expression changes across all four GB subtypes are associated to synapse organization, glutamatergic synapses and chromatin binding

Transcriptome profiles were obtained for each of the 15 patient-derived cell lines, normal human astrocytes and OPCs by bulk RNA-seq. To identify gene expression changes across all four GB subtypes, we first performed a pair-wise differential expression analysis of each GB line *vs* the control astrocytes (Fig. 1c, Supplementary Fig. 2a–c), and then intersected the differentially expressed genes (DEGs) resulting from the 15 pair-wise comparisons (Fig. 1c). We thus identified a set of 497 DEGs that are differentially expressed across all four GB subtypes (Fig. 1d, Supplementary Data 1). Gene Ontology (GO) analysis of this gene set revealed a significant enrichment for terms associated to synapses, and in particular glutamatergic synapses, channel activity and chromatin DNA binding (Fig. 1e). Other GO terms enriched were those related to morphogenesis and development, processes well known to be dysregulated in cancer. A closer analysis of the 497 DEGs showed that 124 genes are annotated as DNA binding, 76 genes encode for Transcription Factors (TFs) and additional 7 genes encode chromatin remodellers (Fig. 1f), evidencing common changes in chromatin-related genes and TFs across the four GB subtypes. Similar findings were obtained following the same approach to compare the GB lines to OPCs, where we identified 2071 DEGs (Supplementary Figs. 3a–c and 4a, Supplementary Data 2) that were also enriched for GO terms related to synapse organization and axon guidance (Supplementary Fig. 4b), or have roles in transcriptional regulation and chromatin remodelling (Fig. 1g). Taken together, 130 genes were differentially expressed in the 15 GB lines independent of the control line used for comparison (astrocytes or OPCs) (Fig. 1h, i). Among these are a relevant fraction that are associated to different neural-related functions (Fig. 1j, k). Altogether this data shows that common gene expression

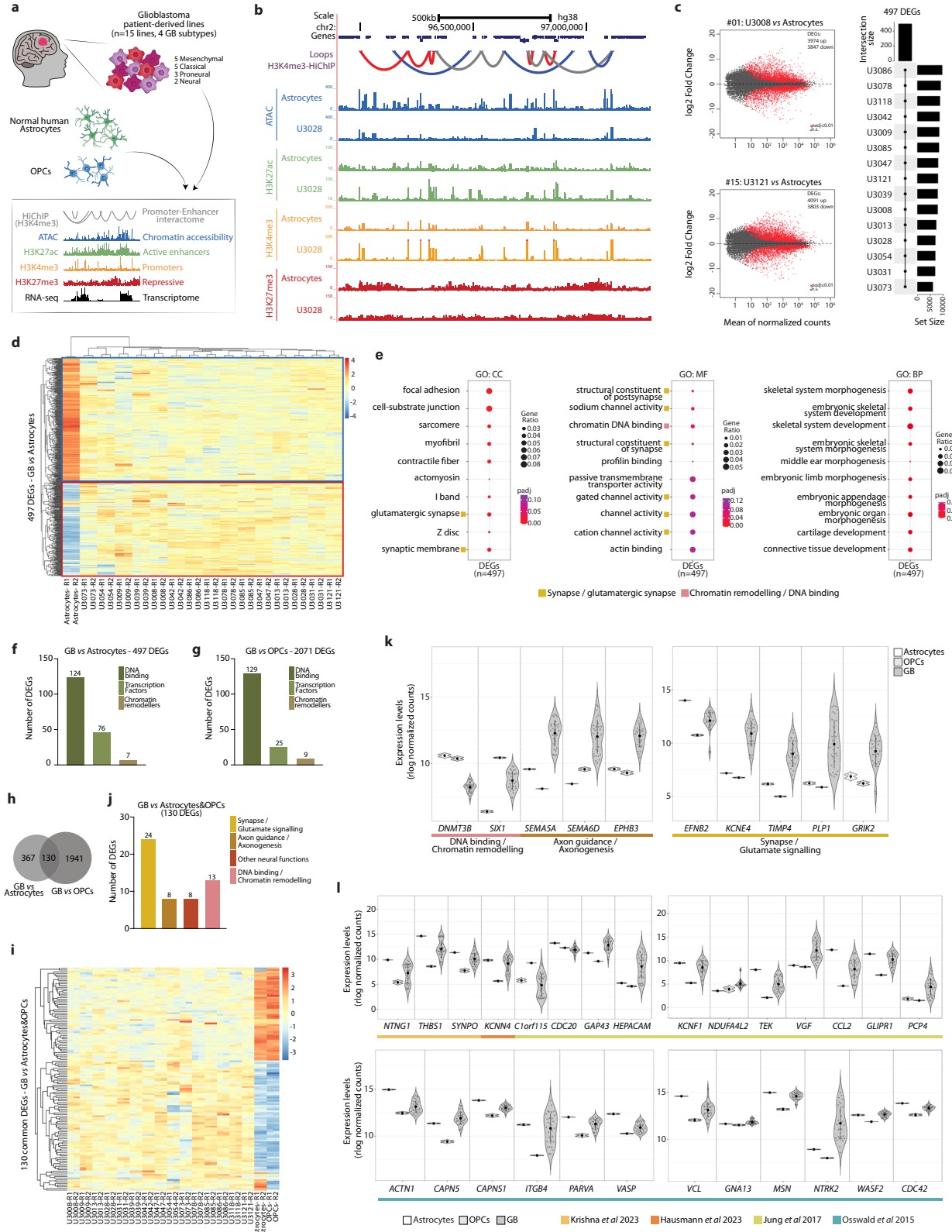

changes in GB are associated to synapse organization, glutamatergic synapses and chromatin processes.

Noteworthy, we also detected changes in the expression of genes underlined in recent literature for their relevance in GB (Fig. 1l). Among those are *THBS1*, encoding thrombospondin-1, involved in the assembly of neural circuits[44]; *KCNN4*, encoding the potassium channel KCa3.1, relevant for autonomous rhythmic activity in glioma networks[45]; and several other genes crucial for the formation of tumour microtubes of high importance for receiving neurogliomal synaptic input[46,47].

In addition, we detected expression of neurotransmitter receptor genes in all 15 GB lines (Fig. 2a), including expression of AMPA, kainate, NMDA and mGluR, which are receptors that respond to glutamate, and in accordance with the glutamatergic identity

**Fig. 1 | Gene expression changes common to all four GB subtypes relate to synapse organization, glutamatergic synapses and chromatin binding.**
**a** Experimental workflow (OPCs: oligodendrocyte progenitor cells). **b** Genomic distribution of HiChIP loops, chromatin accessible regions by ATAC, and H3K27ac, H3K4me3 and H3K27me3 peaks in normal astrocytes and one of the 15 GB cell lines (U3028). **c** Differentially expressed genes (DEGs, red dots) in two representative GB lines *vs* normal astrocytes (left; *p*-value: two-sided Wald's likelihood test with Benjamini correction, *p* < 0.01 and FDR < 0.01) and intersection of the 15 pairwise differential expression analysis resulting in 497 DEGs (right). **d** Expression of 497 DEGs in all 15 GB lines and normal astrocytes as determined by RNA-seq (rlog normalized counts). **e** Top 10 Gene Ontology terms enriched in the 497 DEGs

(*p*-value: hypergeometric test with Benjamini correction). **f, g** Number of DEGs annotated as DNA-binding, transcription factors or chromatin remodellers for the 497 DEGs GB *vs* astrocytes (**f**) and 2071 DEGs GB *vs* OPCs (**g**). **h** Intersection of the DEGs in GB *vs* astrocytes or OPCs yields 130 DEGs in common. **i** Expression of the 130 common DEGs in all 15 GB lines, astrocytes and OPCs (rlog normalized counts). **j, k** Out of the 130 common DEGs, number of genes annotated as synapse/gluta-mate signalling, axon guidance/axonogenesis or DNA binding/chromatin remo-delling (**j**) and expression of selected genes in astrocytes, OPCs and 15 GB lines (**k**) (mean ± SD). **l** Expression of genes with relevance in GB pathogenesis as recently described (refs. 44–47) in astrocytes, OPCs and 15 GB lines (mean ± SD). Source Data are provided as a Source Data file for Fig. 1f, g, j, k, l.

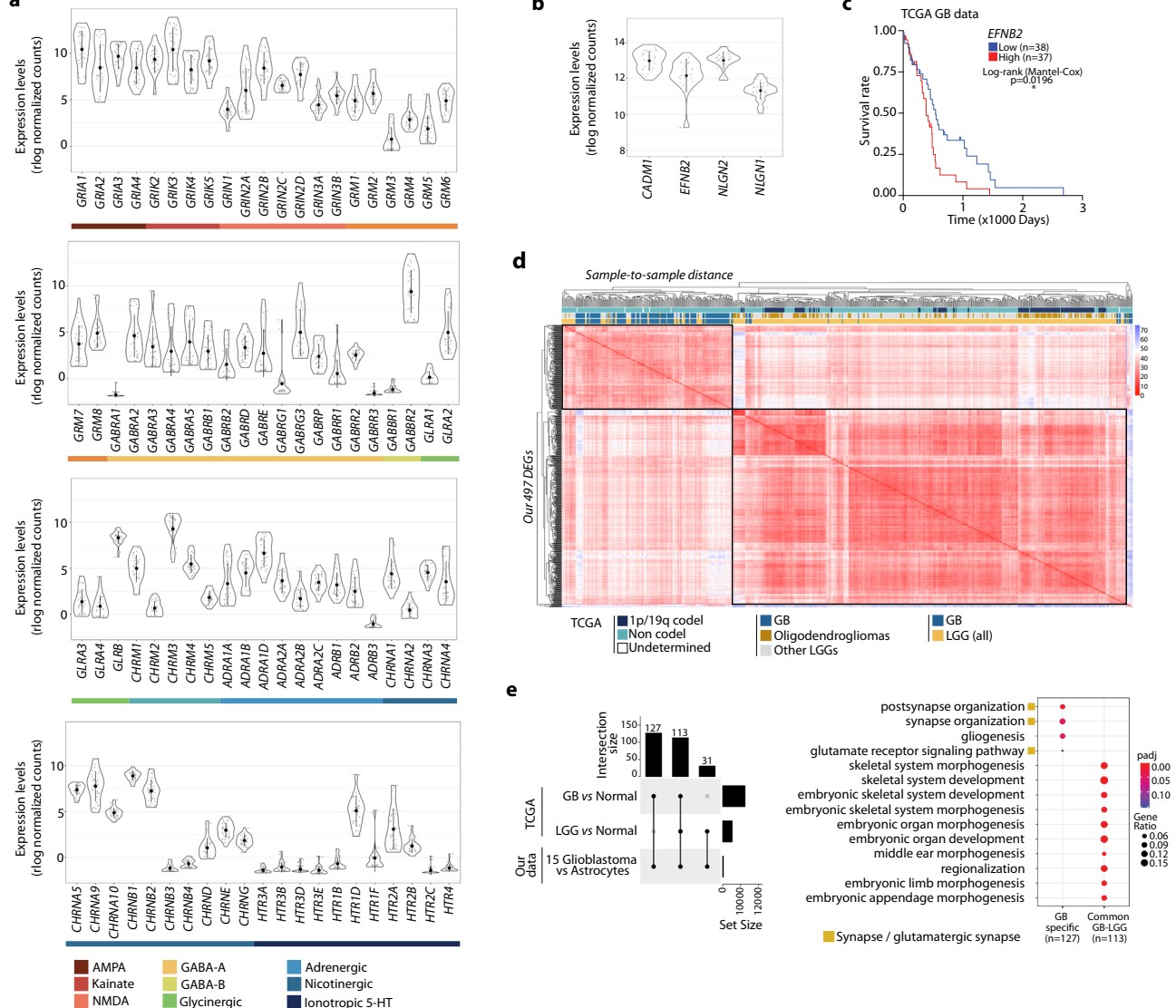

**Fig. 2 | Transcriptional changes reveal a gene signature that segregates GB from LGG. a, b** Expression of neurotransmitter receptor genes (**a**) and synapto-genesis markers (**b**) in the 15 GB lines (mean ± SD). **c** Kaplan–Meier survival curves plotted for patients stratified by quartiles of *EFNB2* expression (TCGA data) (log-rank Mantel–Cox test, *p* = 0.0196, low *n* = 38 and high *n* = 37). **d** Sample-to-sample distance for GB (glioblastoma, *n* = 156) or LGG (low-grade glioma, *n* = 511) tumours from TCGA based on the expression of our 497 DEGs. Annotations include 1p/19q

codeletion and non codel (i.e., 1p/19q intact), as well as LGG oligodendrogliomas and other LGGs. **e** Intersections between our 497 DEGs and the differentially expressed genes from TCGA tumour samples, either GB *vs* normal or LGG *vs* normal tissue (left). Top 10 GO terms enriched in the 127 GB-specific genes or 113 genes common to GB and LGG (right) (*p*-value: hypergeometric test with Benjamini correction).

of the neurogliomal synapses[32]. In particular, the kainate receptor gene *GRIK2* and the NMDA receptor gene *GRIN2C* are also differ-entially expressed across all four GB subtypes. The expression of four known synaptogenesis markers[48] is also detected in our panel

of GB lines (Fig. 2b), in agreement with the emergence of cell subpopulations with synaptogenic properties during glioma progression[49]. Notably, high expression level of the synaptogenesis marker *EFNB2* correlates with lower overall survival of GB

patients (TCGA data, Fig. 2c), evidencing its clinical relevance in glioblastoma.

Importantly, the 497 genes differentially expressed in all 15 GB *vs* astrocytes constitute a gene signature that segregates glioblastoma (GB) from low-grade gliomas (LGG). In a panel of more than 600 tumour samples available at The Cancer Genome Atlas (TCGA), including GB tumours (GB, $n = 156$) and low-grade gliomas (LGG, $n = 511$), the expression of our 497 DEGs can accurately segregate GB from LGG (Fig. 2d). Remarkably, GB samples do not preferentially cluster with 1p/19q codel LGGs and oligodendrogliomas, that correspond to the LGGs with longer overall survival and where no neurogliomal synapses have been detected so far. We then performed a differential expression analysis of the TCGA RNA-seq data from either GB or LGG samples with respect to the corresponding available normal controls, followed by intersection with the 497 DEGs from our data. As a result, we obtained a subset of 127 GB-specific genes and 113 genes "common to all gliomas", the latter resulting from the intersection of both GB gene sets and the LGG gene set (Fig. 2e). Interestingly, GO analysis demonstrated that the 113 genes that are common to all gliomas are preferentially related to morphogenetic and developmental processes, whereas the subset of 127 GB-specific genes is very specifically associated to synapse organization, gliogenesis and glutamate receptor signalling (Fig. 2e). The neurogliomal synapses (i.e., bona fide synapses between presynaptic neurons and postsynaptic glioma cells) utilize glutamate receptors and trigger postsynaptic signals, which in turn affect proliferation and migration of the tumour cells[32–34]. Our analysis reveals that gene expression changes affecting synapses, in particular glutamate receptor signalling, are associated more strongly with glioblastoma samples when compared with lower grade gliomas. This points to certain genes that may be predominant regulators of neural function in GB pathogenesis.

### Loss of long-range regulatory interactions and gain of promoter hub interactions characterize the 3D organization of the GB genome

Control of transcription is exerted by the physical interaction between enhancers and promoters through a non-linear relationship[50]. To chart these physical interactions genome-wide in GB, we performed HiChIP with an antibody against the promoter mark H3K4me3. We thus obtained a map of the promoter interactome in our panel of GB lines representing all four subtypes, including not only promoter-enhancer (P-E) but also promoter-promoter (P-P) and enhancer-enhancer (E-E) interactions (Fig. 3a). A comparison of the loops detected by H3K4me3-HiChIP shows that 4316 loops detected in the normal astrocytes are also present in the GB samples, while an additional 5633 are differential loops: 2125 loops are lost (i.e., astrocyte-specific) and 3508 loops are gained in glioblastoma (i.e., GB-specific) (Fig. 3b). Interestingly, 86.6% of the lost loops involve enhancer interactions (P-E and E-E), while 85.4% of the gained loops involve exclusively promoter-promoter (P-P) interactions (Fig. 3c). Analysis of only multi-anchor loops (i.e., more than two anchor sites, 78% of total loops) shows that a major fraction of these are gained multi-anchor P-P loops (87.4%, Supplementary Fig. 5a), suggesting that gained P-P interactions in GB occur mainly in promoter hubs. Importantly, the length of the lost loops (median ~254 kb) is significantly higher than that of the gained loops (median ~114 kb, $p$-value $= 1.36e^{-290}$), supporting a preferential loss of long-range interactions in GB (Fig. 3d). These changes in the promoter-enhancer interactome are accompanied by gene expression changes: 183 genes located at the anchors of differential loops are differentially expressed across the 15 patient-derived GB lines (Supplementary Fig. 5b), out of which 55 (29.7%) encode for transcription factors, chromatin remodellers and other DNA-binding proteins (Fig. 3e). Our map of the enhancer-promoter interactome reveals topological changes that include loss of long-range

regulatory interactions and gain of promoter hub interactions in all four GB subtypes.

### Remodelling of the regulatory landscape in GB is characterized by loss of regulatory elements and activation of promoters

Depending on the combination of histone marks and other chromatin features, enhancers can present different states from active to silenced, primed or poised[4–8]. To map the regulatory landscape in GB, we profiled by ChIP-seq H3K27ac (active enhancers), H3K27me3 (polycomb-repressed) and H3K4me3 (promoters and enhancers), together with chromatin accessibility by ATAC-seq (Fig. 3f, Supplementary Fig. 6a, Supplementary Data 3). Multiinter intersection of the peaks in the 15 GB lines *vs* the control astrocytes, reveals a redistribution of histone marks and changes in chromatin accessibility occurring across the four GB subtypes (Fig. 3g). This is evidenced by the loss of peaks that were present in astrocytes (i.e., "lost regions") and the remobilization of histone marks to new genomic positions (i.e., "gained regions") in GB. Similarly, a fraction of the chromatin accessible regions defined by ATAC are altered in GB. Moreover, the regions detected in OPCs overlap to a higher extent with those present in astrocytes than in GB (Supplementary Fig. 6b). Altogether, analysis of genomic annotation of the differential regions shows that the remobilization of histone marks results in a loss of active marks at distant elements and an accumulation at gene promoters, while repressive marks are lost from intergenic regions (Fig. 3h, i).

As part of the integration of our multi-omics data genome-wide (Fig. 4a), we used ChromHMM to integrate our datasets and we characterized eight different chromatin states in the normal human astrocytes (Fig. 4b). Plotting the signal of the histone marks and ATAC-seq around the astrocytes' chromatin states displayed clear differences between the GB lines and the control astrocytes (Fig. 4c–e). H3K27ac signal increases in poised and weak promoters but decreases around both strong and weak enhancers (Fig. 4d), further supporting the activation of poised/weak promoters and the loss of regulatory activity at enhancers. In addition, the repressive mark H3K27me3 decreases at poised promoters and polycomb-repressed states in GB (Fig. 4e). These chromatin changes in GB are also accompanied by increased expression levels at inactive/poised promoters and in the proximity of polycomb-repressed regions (Fig. 4f). Such rewiring of the regulatory landscape, together with the changes in the promoter-enhancer interactome and gene expression levels, supports a loss of long-range regulatory interactions and overall activation of hub promoters across all four GB subtypes.

### Remobilization of active chromatin marks around genes associated to glutamatergic synapses, axon guidance and chromatin remodelling

The rewiring of the enhancer landscape in GB leads to increased chromatin accessibility and accumulation of active histone marks at gene promoters. Gene Ontology analysis revealed that those genes near the newly occupied active positions (i.e., gained H3K27ac and ATAC regions) are associated to glutamatergic synapses, axon guidance and axonogenesis, as well as chromatin remodelling/DNA binding (Fig. 5a). Importantly, a major fraction of the genes associated to these biological processes are differentially expressed across the four GB subtypes (Fig. 5b, Supplementary Fig. 7a). Moreover, the loss of repression (i.e., lost H3K27me3 regions) occurs in the vicinity of genes related to calcium ion homeostasis, ion transport and synaptic transmission (Supplementary Fig. 7b). We therefore observed an accumulation of active marks around genes related to synapses, axon guidance and axonogenesis, and loss of repression of other genes associated to ion transport and synaptic transmission. Altogether, this suggests that the rewiring of the regulatory landscape in GB orchestrates a series of gene expression changes that contribute to the synaptic communication between neurons and glioma cells.

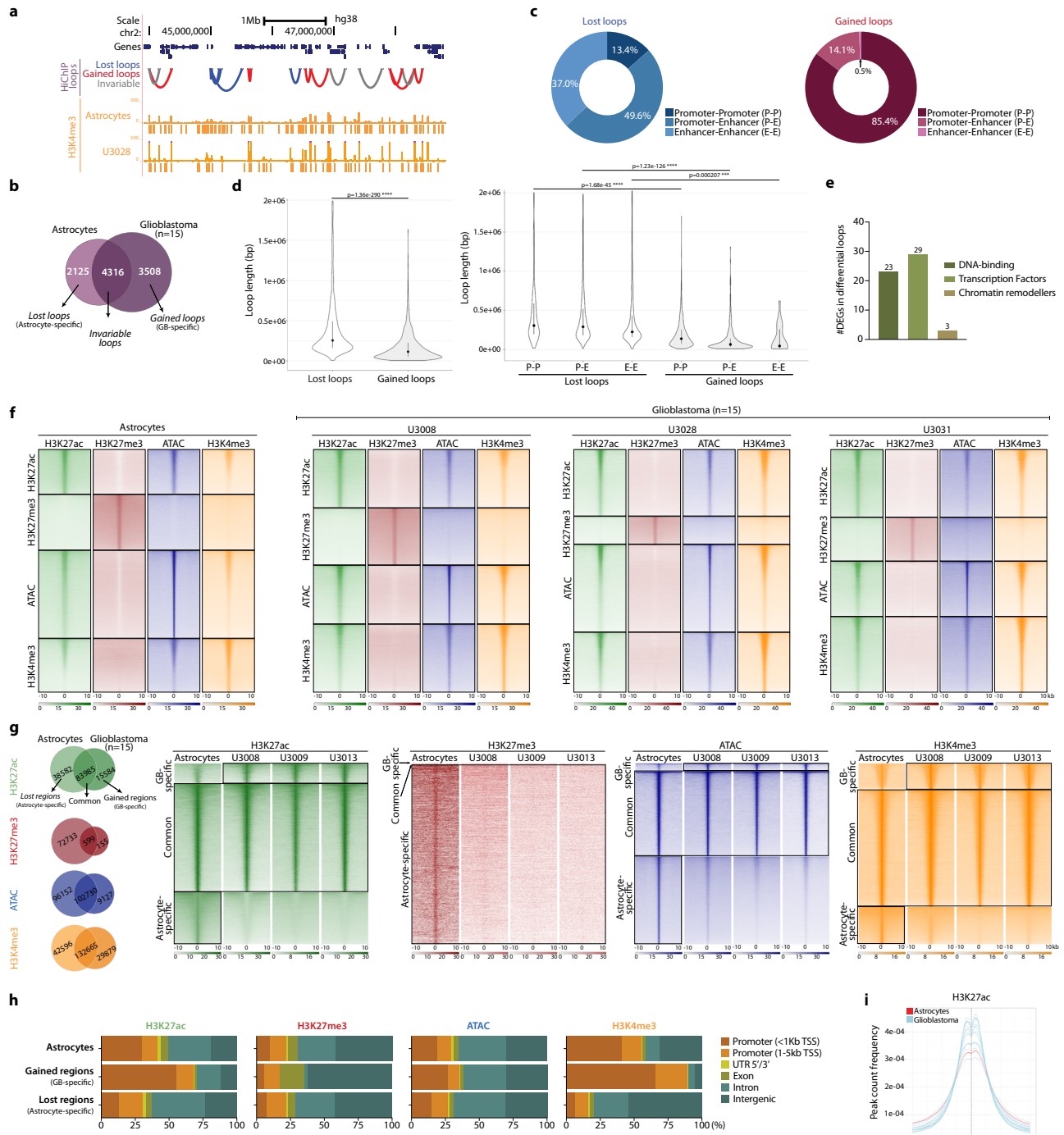

**Fig. 3 | Loss of long-range regulatory interactions and changes in the regulatory landscape in GB. a** Distribution of HiChIP loops and H3K4me3 peaks in a region of chromosome 2. **b** Intersection of HiChIP loops in the 14 GB lines and normal astrocytes defines lost loops ("astrocyte-specific") and gained loops ("GB-specific") in glioblastoma. **c** Annotation of lost and gained loops as P–P, P–E, or E–E loops. **d** Length (bp) of lost and gained loops (median, IQR; two-sided *t*-test with Benjamini correction; *n* = 2125 lost and *n* = 3508 gained loops). **e** Number of DEGs at

the anchors of differential loops annotated as DNA-binding, transcription factors or chromatin remodellers. **f** Read distribution of histone marks and ATAC around the indicated ±10 kb regions in astrocytes and three representative GB lines.
**g** Redistribution of histone marks and ATAC regions in GB. **h** Genomic annotation of differential histone peaks and ATAC regions in GB *vs* normal astrocytes.
**i** H3K27ac peak count frequency around the TSSs. Source Data are provided as a Source Data file for Fig. 3**c**, **d**, **e**, **h**.

## SMAD3 and PITX1 regulatory networks control synapse organization and axonogenesis in GB

A search for transcription factor binding site (TFBS) motifs revealed the enrichment of 11 key TFBS motifs within the newly occupied active regions in GB (Fig. 5c, Supplementary Fig. 7c, d). Not only are the motifs of these 11 TFs enriched in the gained active and open regions, but the genes encoding these 11 TFs are also differentially expressed

across all four GB subtypes (i.e., 11 TFs out of the 76 TFs differentially expressed, Fig. 1f). Moreover, a significant fraction of their direct downstream target genes (i.e., motif within a gained peak <2 kb TSS) are also differentially expressed in the four GB subtypes (Fig. 5d, Supplementary Fig. 8a, b). These findings reflect the impact that chromatin mark redistribution and accessibility changes around these TFBS have on gene expression.

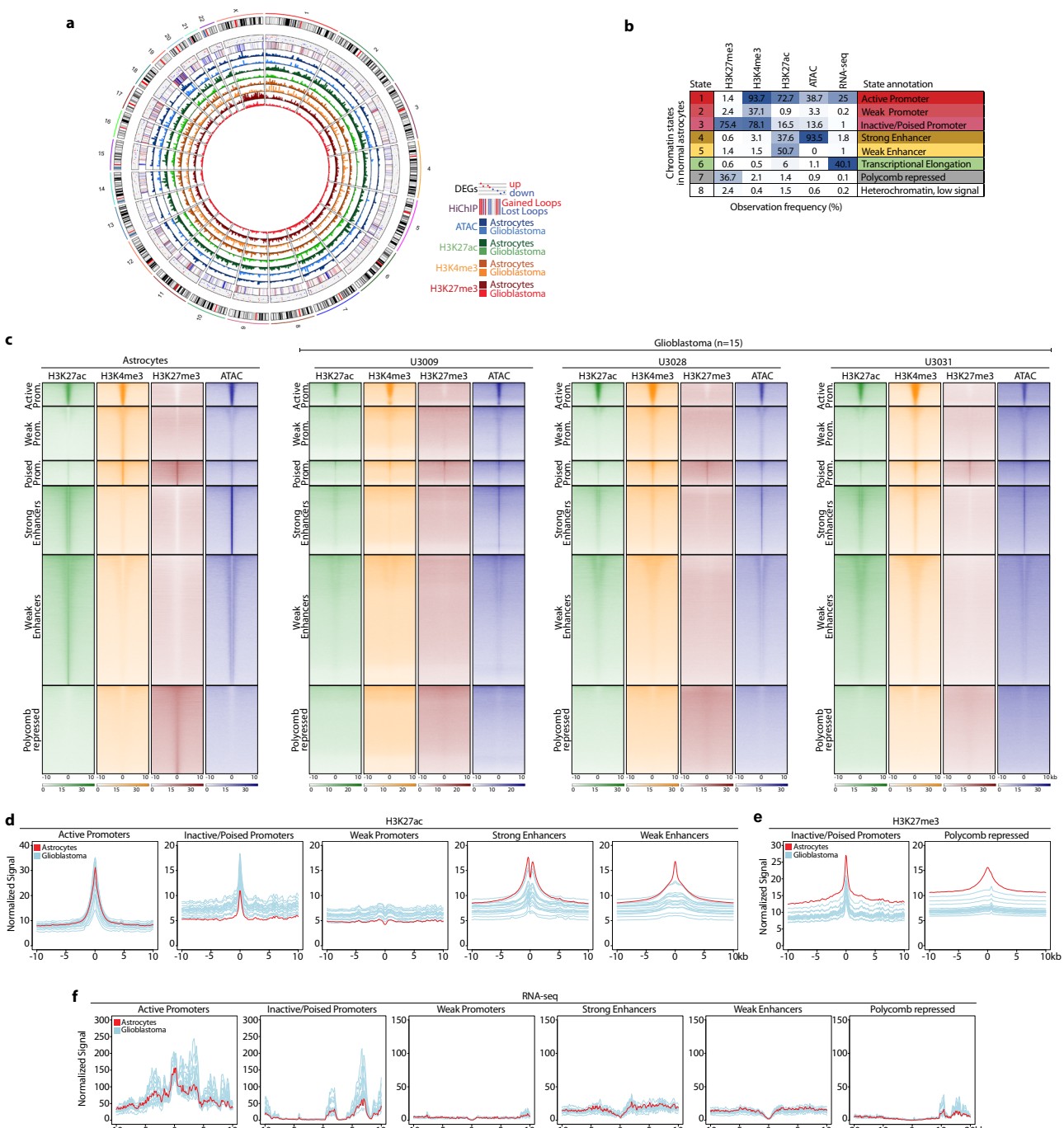

**Fig. 4 | Remodelling of the regulatory landscape in GB includes reduction of active marks at enhancers and activation of promoters. a** Genome-wide integration of all the multi-omics data. **b** Chromatin states defined by chromHMM in normal astrocytes. **c** Read distribution of histone marks and ATAC across the various chromatin states in astrocytes and three representative GB lines. **d**–**f** Density plots representing the normalized H3K27ac (**d**), H3K27me3 (**e**) and RNA-seq (**f**) signal around the astrocytes' chromatin states in glioblastoma (blue) *vs* astrocytes (red).

GO analysis of the downstream target genes of these 11 TFs points to SMAD3 and PITX1 as major players in the regulatory networks that mediate neurogliomal synaptic communication (Fig. 5e, Supplementary Figs. 8c, d and 9). Various GO terms related to synapse density, postsynaptic organization, axon guidance and axonogenesis are significantly enriched in the case of SMAD3 downstream targets, and in particular glutamatergic synapses for the PITX1 downstream target genes (Fig. 5e). This is in agreement with the reported glutamatergic identity of the neurogliomal synapses[32]. GO terms related to TGF-β receptor activity are significantly enriched among the SIX1 downstream targets (Fig. 5e), which links TGF-β signalling via SMAD3 to the synaptic communication between neurons and glioma cells. In addition, the expression of the SMAD3 and PITX1 downstream targets can accurately segregate GB from LGG in a panel of >600 TCGA tumour samples (Fig. 5f). Importantly, 70.2% of the SMAD3 target genes differentially expressed in GB (i.e., 33 out of 47) are also PITX1 downstream targets (Fig. 5d, in bold). The common SMAD3/PITX1 targets include genes encoding semaphorins and ephrins involved in axon guidance[51,52] such as *SEMA5A* and *EPHB3*, the latter of which is also known to participate in the development of excitatory synapses[53]. Other SMAD3/PITX1 targets include the transcription factor *NKX2-2* that is involved in regulating axon guidance[54]; *HCN2* and *KCNE4* that

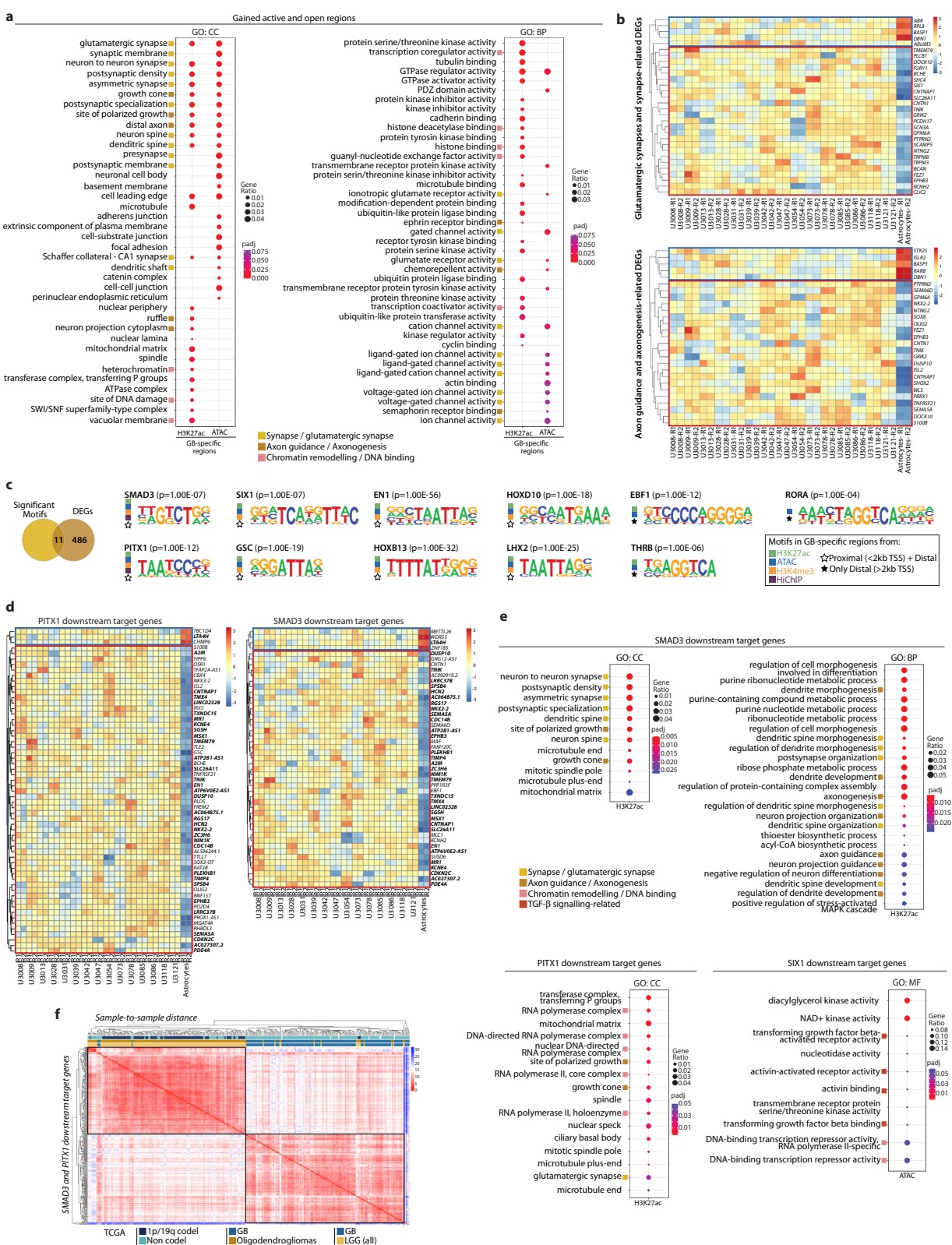

encode for gated channels, the latter considered to regulate neurotransmitter release[55]; as well as *TNIK* which is implicated in glutamatergic signalling, where it binds to NMDA receptors and is required for AMPA expression and synaptic function[56]. Apart from gene expression changes, we also detected differences in protein levels for SMAD3/PITX1 downstream targets such as TNIK, EPHB3 and KCNE4 in the GB lines in comparison to normal astrocytes (Supplementary Fig. 10a–o).

These and other SMAD3/PITX1 downstream targets also present different expression levels in high-grade glioma (HGG) and low-grade glioma (LGG) tissue samples from the human protein atlas (Supplementary Fig. 11).

Apart from SMAD3/PITX1, additional regulatory networks could contribute to a certain extent to the neural role in GB pathogenesis, since other transcription factors such as EBF1, THRB, EN1, HOXB13 and

**Fig. 5 | SMAD3 and PITX1 regulatory networks control genes associated to (glutamatergic) synapse organization and axonogenesis in GB. a** Top 25 GO terms enriched for genes proximal to gained active and open regions in GB (peaks <2 kb TSS) (*p*-value: hypergeometric test with Benjamini correction). **b** Expression of DEGs related to synapse and glutamatergic synapse (top), and axon guidance and axonogenesis (bottom) located <2 kb from a gained active/open peak. **c** Intersection of the 497 DEGs with the HOMER significantly enriched motifs at the gained regions or anchors of differential loops identifies 11 key TFs (*p*-values: hypergeometric test). **d** Expression of PITX1 and SMAD3 downstream target genes (i.e., motif enrichment at a differential peak <2 kb TSS) among the DEGs in GB *vs* normal astrocytes (common targets in bold). **e** Top GO terms enriched for SMAD3, PITX1 and SIX1 proximal downstream target genes (*p*-value: hypergeometric test with Benjamini correction). **f** Sample-to-sample distance for GB (glioblastoma, *n* = 156) or LGG (low-grade glioma, *n* = 511) tumours from TCGA based on the expression of the SMAD3 and PITX1 downstream targets. Annotations include 1p/19q co-deletion and non codel (i.e., 1p/19q intact), as well as LGG oligodendrogliomas.

HOXD10 are also enriched at gained active regions distal to genes involved in axonogenesis and axon guidance (i.e., motif within a gained peak >2 kb TSS) (Supplementary Figs. 8d and 9). Nonetheless, it is remarkable that only SMAD3 and PITX1 binding motifs are enriched at the anchors of the differential loops in GB (Supplementary Fig. 7e). Interestingly, Smad proteins have been previously reported to bind CTCF sites in a CTCF-dependent manner in flies[57] and SMAD3 interacts with CTCF in mammalian cells[58]. SMAD3 and PITX1 binding motifs are also located in close proximity (median distance = 56 bp, Fig. 6a) at the promoters of the 33 common target genes differentially expressed across the four GB subtypes. Mapping of the SMAD3 and PITX1 binding sites by CUT&RUN shows differential peaks at the promoters of several of their downstream target genes (Fig. 6b, Supplementary Fig. 12a, b, Supplementary Data 3). Moreover, *SMAD3* and *PITX1* knockdown and overexpression experiments in U3013 GB cells indicate changes in the expression of a set of target genes involved in synapses, axon guidance and other neural functions (Fig. 6c, Supplementary Fig. 12c–h).

Comparison of *SMAD3-PITX1* and *SMAD3-SIX1* co-expression showed a significant negative correlation between their expression levels, both in our panel of 15 patient-derived samples (Fig. 6d) and in 667 TCGA glioma samples (Fig. 6e). While *SMAD3* is downregulated, *PITX1* and *SIX1* are upregulated in GB (Figs. 1d, 6d, Supplementary Figs. 10a–d, i–j and 11). Moreover, low *SMAD3* and high *PITX1* expression levels correlate with lower overall survival in patients (TCGA data, Fig. 6f), pointing to their clinical relevance. It is important to highlight that both the *SMAD3* and *PITX1* loci present topological and regulatory changes (i.e., differential loops, redistribution of chromatin marks and chromatin accessibility) in comparison to normal astrocytes (Fig. 6g). Also, both *SMAD3* and *PITX1* genes are differentially expressed in all four GB subtypes, and their respective TFBSs are enriched at the promoters of genes related to synaptic function, axon guidance and axonogenesis. This altogether indicates the presence of a regulatory network involving SMAD3 and PITX1, and it suggests that they may be the most prominent TFs regulating the neurogliomal synaptic interaction and axonogenesis in GB.

### SMAD3 inhibition and neuronal activity stimulation cooperate to promote cell proliferation in GB

To functionally support the role of TGF-β signalling and in particular SMAD3 in this context, we tested the effect of their inhibition on the proliferation of GB cells. First, we induced the reprogramming of glutamatergic neurons (ab259259) from human iPSCs (induced pluripotent stem cells) (Fig. 7a–c), and then established co-cultures of glutamatergic neurons and the GB line U251-GFP at different time points (Fig. 7a, d). The U251-GFP line was selected to validate our observations independently, in a cell line other than the 15 GB lines used in the multi-omics approach. By live-cell imaging, we determined the proliferation of the GB cells in co-culture with glutamatergic neurons upon stimulation of neuronal activity alone or in combination with inhibition of either SMAD3 or the TGF-β receptor ALK5. Treatment with the SMAD3-specific inhibitor SIS3 significantly increases the proliferation of GB cells in co-culture with glutamatergic neurons, both at early and late time-points (days 7–10 and 15–18, respectively) (Fig. 7e, f). Stimulation of neuronal activity by picrotoxin has only modest effects on U251 proliferation at the

highest doses, and only with more mature neurons (day 15–18) (Supplementary Fig. 13a). Importantly, combination of SMAD3 inhibition with stimulation of neuronal activity (i.e., SIS3 + picrotoxin) induces proliferation of GB cells to levels significantly higher than those of SIS3-treatment alone (Fig. 7e, f). This effect in proliferation is not observed upon treatment with the ALK5 inhibitor A83 alone; however, the combination of A83 and picrotoxin also leads to increased proliferation compared to untreated cells (Fig. 7g, h), though to a lesser extent than upon SIS3 treatment. Noteworthy, TGF-β signalling inhibition in combination with picrotoxin treatment only promotes proliferation of the GB cells in co-culture with glutamatergic neurons, not when cultured in the absence of neurons (Supplementary Fig. 13b). In line with our multi-omics data, these functional assays suggest that inhibition of SMAD3 and stimulation of neural activity additively cooperate to promote cell proliferation in GB cells in co-culture with glutamatergic neurons. In addition, in vivo inhibition of SMAD3 accelerates the disease progression in GB patient-derived xenografts (PDX) in mice (Fig. 7i–k, Supplementary Fig. 13c, d). Altogether, our data suggests that both in vivo and in vitro SMAD3 inhibition and neural activity cooperate to promote GB progression.

## Discussion

Up to now, few recent reports had integrated chromatin/epigenetic profiling and transcriptomics in glioblastoma[26–29], and much effort had been put on identifying distinct molecular features of GB subtypes[22,23]. This is of great importance to identify clinically relevant subpopulations with the goal of improving the outcome, however it has not yet resulted in clinical benefits. Here, using a broad panel of patient-derived GB cell lines alongside normal human astrocytes and OPCs as controls, we identified changes in the promoter-enhancer interactome, chromatin accessibility and redistribution of histone marks that are present across all four GB expression subtypes (Fig. 8a). Such rewiring of the regulatory landscape and 3D organization of the GB genome orchestrates gene expression changes which underlie neurogliomal synaptic communication. This is manifested by changes in the expression of genes related to synapse organization, axon guidance and axonogenesis, as well as chromatin binding/remodelling. Remarkably, we detected transcriptional changes in genes involved in the assembly of neural circuits[44], autonomous rhythmic activity in glioma networks[45], and neurodevelopmental pathways crucial for the formation of tumour microtubes of high importance for receiving neurogliomal synaptic input[46,47].

Chromatin profiling revealed a preferential loss of long-range regulatory loops and reduction of the active mark H3K27ac at strong and weak enhancers in GB, together with overall activation of promoters, as evidenced by higher enrichment of active marks at poised and weak promoters and reduction of the repressive mark H3K27me3 at those regions. Further supported by CUT&RUN experiments, motif analysis revealed a significant enrichment of SMAD3 and PITX1 sites within gained active and open regions that are located at the promoters of genes related to synapse organization, in particular glutamatergic synapses, axon guidance and axonogenesis. Among the common SMAD3/PITX1 targets, it is worth highlighting genes such as *SEMASA* and *EPHB3*, classical axon guidance molecules[51,52]; the

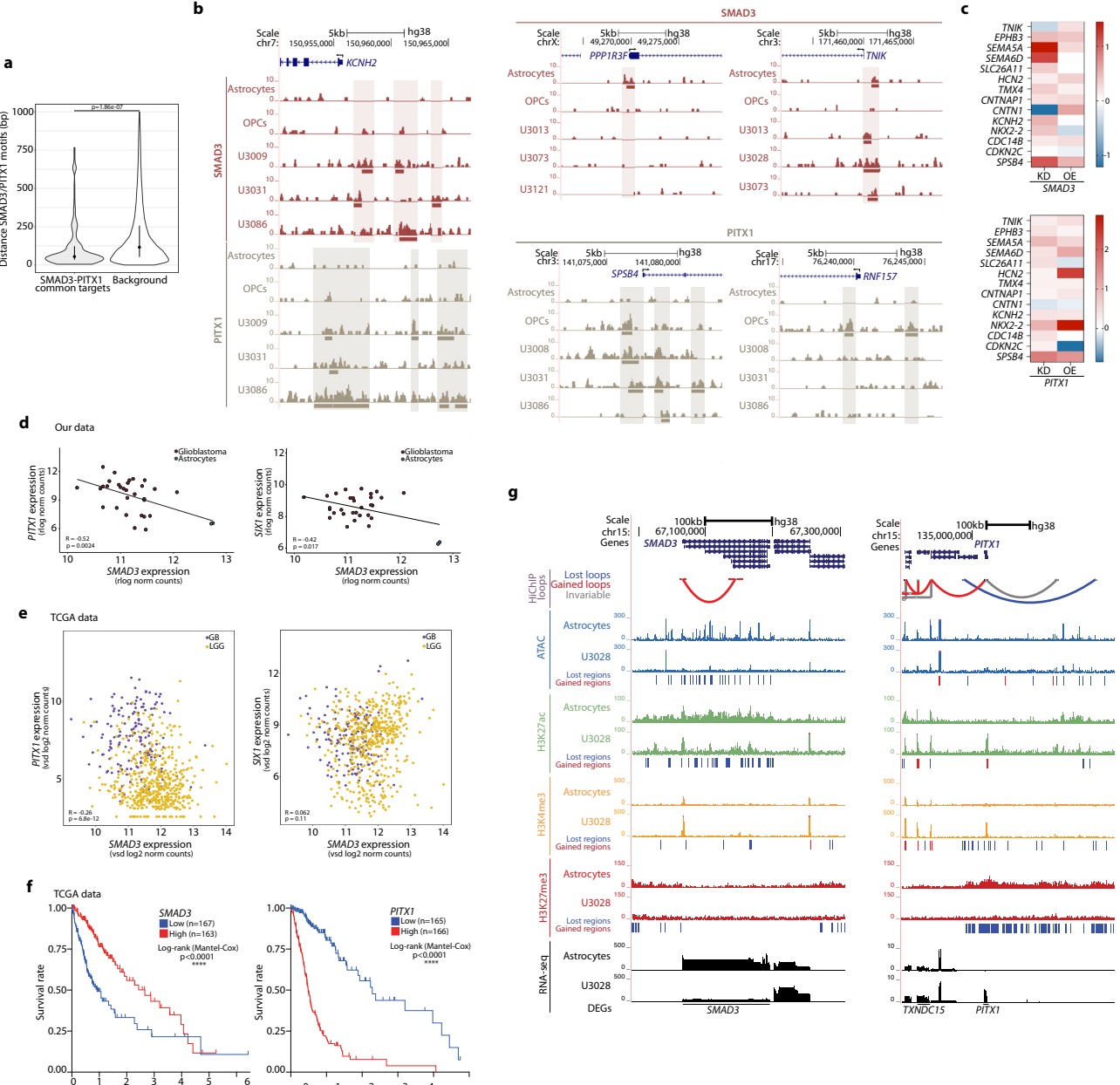

**Fig. 6 | Regulatory and topological alterations in the *SMAD3* and *PITX1* loci accompanied by inversely correlated gene expression and association to lower survival in patients. a** Distance between SMAD3 and PITX1 motifs at the promoters of the 33 common SMAD3/PITX1 target genes (left, median = 56 bp) *vs* background model (right, median = 124 bp) (median, IQR; two-sided Wilcoxon test). **b** Distribution of SMAD3 and PITX1 binding sites at the promoters of representative SMAD3/PITX1 downstream targets in astrocytes, OPCs and three representative GB lines. **c** Changes in the expression of downstream target genes upon knockdown (KD) or overexpression (OE) of *SMAD3* (top) or *PITX1* (bottom) in U3013 cells. Values are represented as $\log_2$FC *versus* the corresponding non-target shRNA or empty vector controls. (two independent experiments and three qPCR technical

replicates per condition). **d, e** Scatterplots showing co-expression of SMAD3-PITX1 (left) or SMAD3-SIX1 (right) expressed as normalized counts in our data (**d**) and TCGA datasets (**e**) (Pearson's correlation coefficients, *p*-value: two-sided correlation test) [two biological replicates in 15 GB lines and astrocytes in (**d**); *n* = 156 (GB) and *n* = 511 (LGG) in (**e**)]. **f** Kaplan–Meier survival curves plotted for patients stratified by quartiles of *SMAD3* (low *n* = 167, high *n* = 163) or *PITX1* expression (low *n* = 165, high *n* = 166) (TCGA data) (log-rank Mantel–Cox test, ****$p$ < 0.0001). **g** Changes in the enhancer landscape and promoter-enhancer interactome at the *SMAD3* and *PITX1* loci (U3028 depicted as representative GB line). Source Data are provided as a Source Data file for Fig. 6**a, c**.

transcription factor *NKX2-2* involved in axon guidance[54]; *HCN2* and *KCNE4* encoding for gated channels; and *TNIK* involved in glutamatergic synaptic function[56]. Moreover, regulatory and topological alterations in the *SMAD3* and *PITX1* loci are accompanied by changes in the expression of both *SMAD3* and *PITX1* genes, which are inversely correlated and associated to lower survival rates in patients. The remaining identified transcription factors (i.e., EBF1, THRB, EN1, HOXB13, HOXD10) could also contribute by regulating distal genes

involved in axonogenesis/axon guidance, and studies using single-cell approaches could in the future reveal additional gene regulatory networks. Even though, our data altogether suggests that SMAD3 and PITX1 act as major direct regulators of a set of downstream target genes related to synapse organization, glutamatergic synapses and axon guidance in GB. Interestingly, SMAD3 and PITX1 motifs are also significantly enriched at the anchors of differential loops, raising the question of whether they have functions directly linked to chromatin

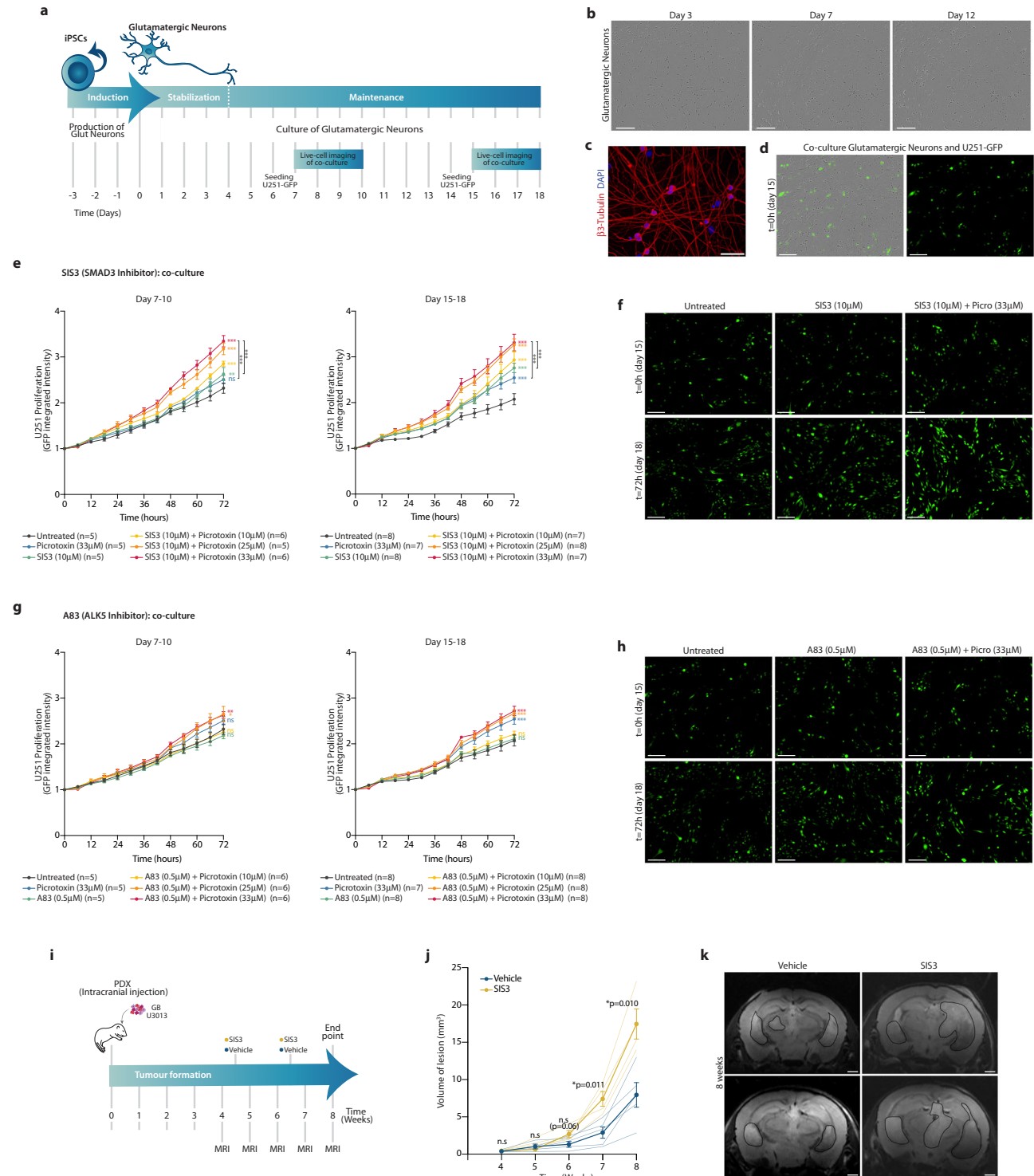

organization. Previous reports have shown that SMAD3 interacts with CTCF in mammalian cells[58] and Smad proteins bind CTCF sites in a CTCF-dependent manner in flies[57]. However, whether PITX1 interacts with CTCF or to what extent SMAD3 and PITX1 contribute to 3D chromatin organization are aspects that remain to be explored.

The recently discovered neurogliomal synapses provide glutamatergic synaptic input that drives tumour progression[32,33], induces formation of tumour microtubes and speeds up tumour cell invasion by hijacking neuronal migration mechanisms[34]. High-grade gliomas can also remodel neural circuits in the human brain promoting tumour progression and decreasing patients' survival[44]. Importantly, using co-

cultures of GB cells and glutamatergic neurons, we functionally demonstrated that inhibition of TGF-β signalling, and specifically SMAD3, in combination with stimulation of neural activity, promotes the proliferation of GB cells. SMADs can act both as transcriptional co-activators and co-repressors via interaction with various transcriptional regulators[59]. Cell type-specific transcription factors[60] and epigenomes[61] can modulate SMAD3 downstream targets and therefore orchestrate cell type-specific effects of TGF-β signalling, a pathway which in cancer has dual roles in the regulation of cell death and proliferation depending on the context[62]. This is noteworthy since glioblastoma cells can transition between states under selective

**Fig. 7 | SMAD3 inhibition and neuronal activity cooperate to promote pro-liferation of GB cells in co-culture with glutamatergic neurons and in vivo in mice. a** Workflow to reprogram glutamatergic neurons from iPSCs, establish a co-culture with GFP-labelled U251 GB cells and live-cell image acquisition. **b** Photomicrograph of glutamatergic neurons at days 3, 7 and 12. **c** Glutamatergic neurons stained with an antibody against β3-Tubulin (red) (DAPI, blue). **d** Photomicrographs of glutamatergic neurons and U251-GFP glioblastoma cells (green) in co-culture. **e**, **g** Proliferation curves depicting the growth of U251 cells (measured as normalized GFP integrated intensity) in co-culture with glutamatergic neurons in the 72 h after seeding at day 6 (left) or day 14 (right), while treated with either the SMAD3-specific inhibitor SIS3 (**e**) or the ALK5 inhibitor A83 (**g**), in combination with increasing concentrations of picrotoxin (mean ± SEM, multiple unpaired two-sided *t*-test, *t* = 72 h, * *p* < 0.01, ** *p* < 0.005, *** *p* < 0.000005, exact *p*-values in Source Data file) [(**e**, left) *n* = 5 in U, P33, S10, S10 + P25 and *n* = 6 in

S10 + P10, S10 + P33; (**e**, right) *n* = 8 in U, S10, S10 + P25 and *n* = 7 in P33, S10 + P10, S10 + P33; (**g**, left) *n* = 5 in U, P33, A0.5 and *n* = 6 in A0.5 + P10, A0.5 + P25, A0.5 + P33; (**g**, right) *n* = 8 in U, A0.5, A0.5 + P10, A0.5 + P25, A0.5 + P33 and *n* = 7 in P33; where U, P, S and A denote untreated, picrotoxin, SIS3 and A83, respectively, and the numbers indicate μM concentration]. **f**, **h** Representative images from the live-cell imaging proliferation assay of U251-GFP cells in co-culture with glutamatergic neurons, in the presence of either SIS3 (**f**) or A83 (**h**). **i** Timeline of longitudinal study to follow tumour progression in vivo in mice carrying PDXs (patient-derived xenografts) and treated with either the SMAD3 inhibitor SIS3 or vehicle. **j** Volume of pathological lesions measured weekly by MRI from *t* = 4 to *t* = 8 weeks in SIS3- or vehicle-treated mice (*n* = 4 and *n* = 5 respectively; mean ± SEM, two-sided Mann Whitney test). **k** Representative MRI images of SIS3- or vehicle-treated mice at *t* = 8 weeks. Scale bars: 200 μm (**b**, **d**, **f**, **h**), 50 μm (**c**), 1 mm (**k**). Source Data are provided as a Source Data file for Fig. 7e, g, j.

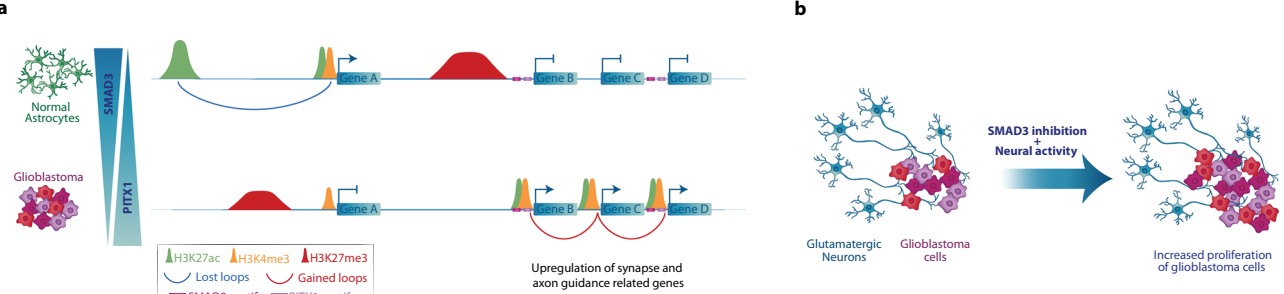

**Fig. 8 | Rewiring of the regulatory landscape and promoter-enhancer inter-actome in GB orchestrates gene expression changes underlying neurogliomal synaptic communication. a** Scheme illustrating the regulatory and topological changes in the GB genome including loss of long-range interactions, gain of promoter-promoter loops, redistribution of active chromatin marks towards

SMAD3 and PITX1 sites and upregulation of genes related to synapse organization, axon guidance and axonogenesis. **b** Illustration depicting that SMAD3 inhibition and stimulation of neural activity cooperate to promote proliferation of GB cells in co-culture with glutamatergic neurons.

pressure (e.g., in response to treatments), which may also result in different effects when comparing different in vivo and in vitro experimental set-ups. Even though SMAD3 inhibition might reduce cell viability of certain GB subtypes[63], our data underlines the importance of the cellular context, i.e., contacts with neurons, where the combination of neural activity stimulation and SMAD3 inhibition have proven to additively cooperate to promote proliferation of GB cells (Fig. 8b), both in co-culture systems that model neuron-to-glioma interactions and in PDX models in mice. Considering these neural-cancer interactions will be pivotal to improve the prognosis of malignancies difficult to treat, such as glioblastoma. Our study thus provides details of the regulatory and topological alterations in GB and offers mechanistic insight into the gene regulatory networks that mediate the neurogliomal synaptic communication.

## Methods

### Ethics approval

Our research complies with all relevant ethical regulations. All experiments with laboratory animals were performed in compliance with national and institutional laws, and according to protocols approved by the Regional Ethics Committee at the Court of Appeal of Northern Norrland (ethical permit ID A29-2019 and A3-2023).

### Cell culture

*Human glioblastoma cell lines* were derived from glioblastoma (GB) patient biopsies and obtained via the Human Glioblastoma Cell Culture (HGCC) resource[35] (Uppsala University, Sweden). The 15 patient-derived GB lines represent the four GB expression subtypes (*n* = 5 classical, *n* = 5 mesenchymal, *n* = 3 pro-neural and *n* = 2 neural). Cells were seeded onto poly-ornithine/laminin-coated plates and grown in Feed Medium [1:1 ratio of DMEM/F12 Glutamax (Gibco) and Neurobasal medium (Gibco), supplemented with 1× B27 (Gibco), 1× N-2

Supplement (Gibco), 1% penicillin/streptomycin (Gibco), 10 ng/ml EGF (Epithelial Growth Factor; PreproTech EC Ltd.) and 10 ng/ml FGF (Fibroblast Growth Factor; PeproTech EC Ltd)].

*Normal Human Astrocytes* (Lonza, CC-2565) were grown in AGM Astrocyte Growth medium BulletKit (Lonza, CC-3186).

The *iNeu™ human oligodendrocyte progenitor cells (OPCs)* (Creative Biolabs NeuroS, #NCL-2103-P49) were seeded onto plates coated with 5% Corning's Matrigel Matrix (#356231, lot: 2284001) and grown in Oligodendrocyte Precursor Cell Growth Medium (#NCL-21P6-105, Creative Biolabs NeuroS) with a medium change regime every 48 h.

*U251 glioblastoma cells* (Sigma-Aldrich, #09063001, authenticated by STR profiling) were grown in EMEM (EBSS) supplemented with 2 mM Glutamine, 1% NEAA (Non-Essential Amino Acids), 1 mM Sodium Pyruvate, 10% FBS (Fetal Bovine Serum) and 1% penicillin/streptomycin (all from Gibco). The GFP-labelled U251 line was established by lentiviral integration of the GFP reporter gene. Given the limitations related to possible genetic drift over time, the U251 line was used only in vitro in co-culture assays to perform functional validations in a line other than the 15 patient-derived GB lines used in the *omics* approaches.

The *iPSC-derived glutamatergic neurons* (ioGlutamatergic neurons) were purchased from Abcam (ab259259), where human iPSCs were exposed to a 3-day induction protocol (day −3 to 0) and ioGlutamatergic neurons were cryopreserved. Upon thawing, 11000 ioGlutamatergic neurons per well were seeded onto poly-D-lysine-Geltrex-coated 96-well plates and grown in Complete Glutamatergic Neuron Medium (CGNM) (i.e., Neurobasal medium (Gibco) supplemented with 1× Glutamax (Gibco), 25 μM 2-mercaptoethanol (Gibco), 1× B27 (Gibco), 10 ng/ml NT3 (R&D) and 5 ng/ml BDNF (R&D)). During the stabilization phase, the CGNM was supplemented with 1 μg/ml doxycycline (Sigma) during 96 h (day 0–4) and DAPT (Sigma) for 48 h (day 2–4) for sustained induction. During the maintenance phase (day 4

onwards), the ioGlutamatergic neurons were grown in CGNM (without doxycycline and DAPT) with a half-medium change regime every 48 h.

To establish the *co-culture of glutamatergic neurons and GB cells*, 5000 U251-GFP GB cells were seeded onto the glutamatergic neurons either at day 6 or day 14 of culture. The co-cultures were treated with Picrotoxin (10, 25 or 33 μM, TOCRIS #1128), SIS3 (10 μM, Calbiochem #566405), A83 (0.5 μM, Sigma SML0788) or combinations of Picro-toxin and SIS3/A83. Live-cell imaging was performed on an IncuCyte S3 Live-Cell Analysis instrument (Sartorius) and proliferation of the GFP-labelled U251 cells was determined by measuring GFP integrated intensity using the Incucyte Base Analysis Software. Data points cor-respond to $n = 5–8$ replicates per condition and timepoint in Fig. 7e, g and $n = 4–8$ in Supplementary Fig. 13a, b, for which 4 fields were imaged, and values are normalized to $t = 0$ (exact $n$ values in figure panel and Source Data file). Significant differences in cell proliferation were assessed using unpaired $t$-test with correction for multiple testing at $t = 72$ h (* $p < 0.01$, ** $p < 0.005$, *** $p < 0.000005$).

*SMAD3* and *PITX1 stable KD (knockdown) and OE (overexpression) cell lines* were established in the patient derived U3013 GB line. KDs were generated using the MISSION® shRNA clones targeting *SMAD3* (clone ID #TRCN0000330128) or *PITX1* (clone ID #TRCN0000415860), and MISSION® TRC2 pLKO.5-puro non-target shRNA control plasmid (#SHC216) (Sigma-Aldrich). For *SMAD3* OE, the pLV-CMV-hSMAD3 plasmid was synthesized by VectorBuilder by cloning the human *SMAD3* coding sequence downstream of a CMV promoter, and the empty pLV vector was used as a control. For *PITX1* OE, the human *PITX1* coding sequence followed by a T2A was cloned immediately upstream the EGFP in the pLenti CMV GFP Puro plasmid (Addgene #17448), and the empty pLenti CMV Puro plasmid was used as a control. All OE constructs contain a CMV-EGFP sequence whose expression enabled subsequent FACS sorting of positively transduced cells. To establish the KD and OE lines, lentiviral particles were gen-erated by transfection of HEK293T cells with the corresponding len-tiviral transfer vector together with the pSPAX and pMD2.G lentiviral packaging plasmids, and using Lipofectamine™ 2000 (Thermo Fisher). 24 h post-transfection, the viral supernatant was filtered and used for transduction of U3013 GB cells. Transduced cells were selected either with puromycin (KD lines) or by FACS sorting (OE lines; GFP-positive cells sorted using the BD FACSAria™ III Cell Sorter instrument and the BD FACSDiva software).

Normal astrocytes, OPCs and patient-derived GB lines were grown in defined media containing the growth factors and components required to preserve their specific cell identity. Cell lines were authenticated as follows: (i) GB patient-derived lines were character-ized in terms of gene expression, copy number variation and patho-logical analysis by the HGCC resource; (ii) normal human astrocytes are provided with a certificate of analysis per lot by Lonza; (iii) iPSC-derived glutamatergic neurons were characterized by RNA-seq at Abcam for the expression of glutamate transporter genes *VGLUT1* and *VGLUT2* and markers *FOXG1* and *TBR1*; (iv) OPCs were characterized morphologically and for the expression of known marker proteins such as O4, PDGFαR, NG2 and CNPase at Creative Biolabs NeuroS; and, (v) U251 cells were authenticated by STR-PCR profiling at Sigma. All cell lines tested negative for mycoplasma using the MycoAlert PLUS detection kit (Lonza, LT07-703), and were grown in a cell incubator at 37 °C in a humidified atmosphere (95% humidity) with 5% $CO_2$.

## RNA-seq

RNA-seq was performed in 15 GB cell lines, normal human astrocytes and human OPCs. Total RNA was extracted using RNeasy Plus Mini Kit (Qiagen, #74134) in duplicates for each cell line. Poly(A) RNA was purified using the NEBNext Poly(A) mRNA Magnetic Isolation Module (CAT #E7490L). RNA-seq libraries were prepared using the NEBNext® Ultra™ II RNA Library Prep Kit for Illumina (CAT# E7770L) and NEB-Next® Multiplex Oligos for Illumina® (96 Unique Dual Index Primer

Pairs) (CAT #E6440S) following the manufacturer's instructions (8 amplification cycles). The RNA-seq libraries were sequenced on a NovaSeq 6000 Sequencing System (Illumina) obtaining in average ~53 million 150PE reads per library.

**RNA-seq analysis.** Fastq files were quality-checked with FastQC (https://www.bioinformatics.babraham.ac.uk/projects/fastqc/, 0.11.8) and raw reads were mapped to the human genome (GRCh38/hg38) using STAR (2.7.6b). Genes with a minimum row sum of 10 reads were kept for further analysis. Normalization and differential expression analysis of each GB line *vs* the control normal astrocytes or OPCs were performed using DESeq2 (1.30.1, 1.38.3, Bioconductor 3.11/3.16) ($p < 0.01$ and FDR < 0.01), and the DEGs resulting from the 15 pair-wise comparisons were then intersected using the UpSet command from the ComplexHeatmap package (2.6.2, 2.14.0), resulting in 497 differ-entially expressed genes (DEGs) across all 15 GB lines *vs* astrocytes and 2071 DEGs across all 15 GB lines *vs* OPCs. Pheatmap package (1.0.12) was used for clustering of differentially expressed genes in Figs. 1d and 5b, d, and Supplementary Figs. 4a, 5b, 7a, 8a, b. Gene Ontology (GO) and KEGG enrichment analysis were performed using clusterProfiler (4.0.4, 4.6.2) ($p < 0.01$ and FDR < 0.01) (CC: cellular component, MF: molecular function, BP: biological process). RNA-seq data available at The Cancer Genome Atlas (TCGA) from both human Glioblastoma (GB, $n = 156$) and Low-Grade Glioma (LGG, $n = 511$) tissue samples were retrieved using the R packages TCGAbiolinks (2.18.0, 2.25.3) and RTCGAToolbox (2.20.0, 2.28.4), together with normal tissue samples (NT, $n = 5$). We calculated Euclidean distances among the tumour samples based on the gene expression of either our 497 DEGs (Fig. 2d) or the SMAD3/PITX1 downstream target genes (Fig. 5f), and plotted the sample-to-sample distances as a heatmap. The differential expression analysis of either GB samples or LGG samples *versus* the respective normal non-tumour control tissues, all retrieved from TCGA, was performed using DESeq2 (FDR < 0.01). The UpSet command from the ComplexHeatmap package was used to intersect the 497 DEGs from our data with the DEGs resulting from the TCGA-GB *vs* Normal and TCGA-LGG *vs* Normal differential expression analysis, and the inter-section was represented as an UpSet plot. Co-expression analysis for *SMAD3-PITX1* and *SMAD3-SIX1* gene pairs and Pearson correlation was calculated and plotted on R and ggplot2 (3.4.3), ggpubr (0.6.0) and rstatix (0.7.2) using the rlog normalized counts for our RNA-seq data and the vsd log2 normalized counts for the TCGA data.

## ChIP-seq

ChIP-seq was performed in 15 GB cell lines, normal human astrocytes and human OPCs as described before[64,65] with some modifications. Briefly, cells were fixed on the plate by adding formaldehyde directly to the medium (final concentration 1% formaldehyde) for 15 min at room temperature while rotating. The crosslinking reaction was quenched by adding Glycine (final concentration 125 mM Glycine) for 5 min, and fixed cells were scraped off and harvested in 1X cold PBS containing protease inhibitors. Cells were then resuspended in lysis buffer ($3–6 × 10^6$ cells/ml) and sonicated in a Covaris E220 instrument (shearing time 12 min, PIP 140, duty factor 5, and 200 cycles per burst), to achieve a fragment size ranging from 200 bp to 700 bp. Chromatin immunoprecipitation was performed with antibodies against H3K27ac (ab4729, 4 μg per ChIP), H3K4me3 (ab8580, 4 μg per ChIP) and H3K27me3 (ab192985, 4 μg per ChIP), and using Dynabeads™ M-280 Sheep Anti-Rabbit IgG (Invitrogen 11203D). H3K27ac and H3K27me3 ChIP–seq libraries were prepared using the NEBNext Ultra II DNA Library Prep Kit for Illumina (E7645L) and NEBNext® Multiplex Oligos for Illumina® (E6440S). The H3K4me3 library was prepared using Accel NGS 2S Plus DNA Library Prep (#21024, Swift Biosciences) and indexing Kit (#26596, Swift Biosciences), since it was processed and sequenced together with the H3K4me3-HiChIP. ~10 ng of immuno-precipitated chromatin (as quantitated by the Qubit fluorometer

alongside the corresponding inputs were amplified for 8 cycles and further processed according to the guidelines of the library prep kits. SPRI Select beads (Beckman Coulter) were used for clean-up and size selection. The ChIP-seq libraries were sequenced on a NovaSeq 6000 Sequencing System (Illumina) obtaining in average ~53 million 150PE reads per library.

**ChIP-seq analysis.** Fastq files were quality-checked with FastQC (https://www.bioinformatics.babraham.ac.uk/projects/fastqc/, 0.11.8) and raw reads were mapped to the human genome (GRCh38/hg38) using bowtie2 (2.4.1) (--threads 4 --very-sensitive). Peak calling was performed by MACS2 (2.2.6) (options: --broad -g hs -B -q 0.05 -f BAMPE) using the corresponding input track as control (i.e., astrocytes *vs* astrocyte input, OPCs *vs* OPCs input, and each of the GB cells *vs* the corresponding input depending on the subtype i.e., CL-input, MS-input, PN-input or NL-input). ChIP-seq signal was plotted as heatmaps using plotHeatmap from deepTools (2.4.3, 2.5.1, 3.1.0, 3.3.2). Bigwig files were visualized in the UCSC genome browser (https://genome.ucsc.edu/). Density plots displaying the signal around TSSs or defined chromatin states were performed with plotProfile upon calculation of enrichment using computeMatrix. Bedtools multiinter tool (2.30.0) was used to determine the regions that were lost or gained in glioblastoma for each of the histone marks. Lost regions were defined as astrocyte-specific regions (i.e., present in astrocytes and absent in all 15 GB lines), while gained regions were defined as GB-specific (i.e., absent in astrocytes and present in at least 10 out of 15 GB lines). Genomic annotations of peaks as well as lost and gained regions were obtained using BSgenome.Hsapiens.UCSC.hg38 (1.4.3, 1.44), ChIPpeakAnno (1.4.3, 1.44), ChIPseeker[66] (1.28.3, 1.34.1) and EnsDb.Hsapiens.v86 (2.99.0) in R. Gene Ontology analysis of genes proximal (<2 kb TSS) or distal (>2 kb TSS) to the gained/lost regions were performed using clusterProfiler (4.0.5, 4.6.2) (CC: cellular component, MF: molecular function, BP: biological process). Search for TFBS (transcription factor binding sites) motifs was conducted using HOMER (v4.11) tool findMotifsGenome.pl. For each histone mark, motif analysis was performed within the differential regions located either proximally (<2 kb) or distally (>2 kb) to TSSs. Significantly enriched TF motifs ($p < 0.01$) were intersected with the 497 DEGs to identify transcription factors that were differentially expressed and whose motifs were enriched at the differential histone regions. Distances between SMAD3 and PITX1 motifs at the promoters of the 33 common SMAD3/PITX1 target genes were calculated within the differential regions (i.e., gained H3K27ac and ATAC peaks) located <2 kb of TSS, and using as background model the common peaks at promoters genome-wide. Discovery of chromatin states in normal astrocytes was performed with ChromHMM[67] (1.23) using the H3K27ac, H3K4me3, H3K27me3, ATAC-seq and RNA-seq datasets as input and setting 8 state emissions.

**ATAC-seq**
ATAC-seq was performed in 15 GB cell lines, normal human astrocytes and human OPCs (50000 cells/sample) as previously described[43]. Briefly, DNA tagmentation was performed 30 min at 37 °C using the Illumina Tagment DNA TDE1 Enzyme and Buffer Kits (#20034197). The reaction was purified using a MinElute Purification Kit (Qiagen #28004) and fragmentation was assessed via Bioanalyzer High Sensitivity DNA Analysis (Agilent # 5067-4626). 5 µl of tagmented DNA per library were amplified for 13 cycles using NEBNext High-Fidelity 2× PCR Master Mix (M0541S) and custom oligonucleotides (for oligo sequences see Supplementary Data 4). The ATAC libraries were sequenced on a NovaSeq 6000 Sequencing System (Illumina) aiming ~55 million 150PE reads per library.

**ATAC-seq analysis.** ATAC-seq analysis was performed according to the ENCODE ATAC-seq Processing Pipeline with some modifications. Fastq files were quality-checked with FastQC (https://www.

bioinformatics.babraham.ac.uk/projects/fastqc/, 0.11.8). The pipeline for further processing included trimming with cutadapt to remove the Nextera adaptor sequence CTGTCTCTTATACACATCT, mapping to the human genome (GRCh38/hg38) using bowtie2 (--k 2, --threads 8, --local, --maxins 2000), removing duplicates with MarkDuplicates from Picard toolbox (2.27.5) and filtering with samtools (1.12) to keep high quality and uniquely aligned read pairs. Peak calling was performed using MACS2 (2.2.6) (--broad -q 0.05 --shift 100 --extSize 200 against baseline). Genomic annotations of peaks were obtained using ChIPseeker[66] (1.28.3, 1.34.1); the ATAC-seq signal was plotted as heatmaps using plotHeatmap from deepTools (2.4.3), and bigwig files were visualized in the UCSC genome browser (https://genome.ucsc.edu/). Density plots displaying the signal around defined chromatin states were performed with plotProfile upon calculation of enrichment using computeMatrix (deepTools- 2.4.3, 2.5.1, 3.1.0, 3.3.2). Bedtools multi-inter tool (2.30.0) was used to determine the ATAC regions that were lost or gained in glioblastoma. Lost regions were defined as astrocyte-specific regions (i.e., present in astrocytes and absent in all 15 GB lines), while gained regions were defined as GB-specific (i.e., absent in astrocytes and present in at least 10 out of 15 GB lines). Lost and gained ATAC regions were then annotated using BSgenome.Hsapiens.UCSC.hg38 (1.4.3, 1.44), ChIPpeakAnno (3.26.4, 3.32.0), ChIPseeker[66] (1.28.3, 1.34.1) and EnsDb.Hsapiens.v86 (2.99.0) in R. Gene Ontology analysis of genes proximal (<2 kb TSS) or distal (>2 kb TSS) to the gained/lost ATAC regions were performed using cluster-Profiler (CC: cellular component, MF: molecular function, BP: biological process). Search for TFBS (transcription factor binding sites) motifs was conducted using HOMER (v4.11) tool findMotifsGenome.pl. Motif analysis was performed within the differential ATAC regions located either proximally (<2 kb) or distally (>2 kb) to TSSs. Significantly enriched TF motifs ($p < 0.01$) were intersected with the 497 DEGs to identify transcription factors that were differentially expressed and whose motifs were enriched at the differential ATAC regions.

**HiChIP**
HiChIP was performed in 15 GB cell lines and normal human astrocytes using the Arima-HiC+ Kit (A101020) and following the guidelines in the Arima-HiChIP user guide for mammalian cells. $9-15 \times 10^6$ cells per line were used to obtain at least 12–15 µg of input DNA for HiChIP. Cells were fixed in 1% formaldehyde for 15 min at room temperature while rotating, and crosslinking reaction was quenched by adding Glycine (final concentration 125 mM Glycine) for 5 min. Subsequent steps included digestion of crosslinked chromatin with restriction enzymes, end-filling with biotinylated nucleotides and ligation. Proximally liga-ted chromatin was then sheared on a Covaris E22O instrument (shearing time 5 min, PIP 105, duty factor 5, and 200 cycles), to achieve a fragment size ranging from 200 bp to 800 bp. Chromatin immuno-precipitation was performed with an antibody against H3K4me3 (ab858, 4 µg per ChIP). After biotin enrichment and adapters ligation, immunoprecipitated DNA was subjected to PCR amplification (8–11 cycles) using Accel-NGS 2 S Plus DNA Library Kit (#21024, Swift Bios-ciences) and indexing Kit (#26696, Swift Biosciences), according to the Arima-HiChIP Library Prep user guide. Quality controls for chromatin digestion, ligation, shearing and library preparation were assessed via Bioanalyzer High Sensitivity DNA Analysis (Agilent # 5067-4626) and passed prior to sequencing. The HiChIP libraries were sequenced on a NovaSeq 6000 Sequencing System (Illumina) aiming ~100 million 150PE reads per library.

**HiChIP analysis.** HiChIP analysis was performed as described before[42] using HiC-Pro and Hichipper. Fastq files were quality-checked with FastQC (https:// www.bioinformatics.babraham.ac.uk/ projects/fastqc/, 0.11.8). Mapping to the human genome (GRCh38/hg38) and retrieval of valid interacting fragments was performed using the HiC-Pro software v3.1.0 and setting ligation sites as

GATCGATC, GANTGATC, GANTANTC, GATCANTC. Valid loops were identified using Hichipper v.0.7.3. Significant loops were determined using diffloop (1.20.0)[68] in R filtering for a minimum normalized read pair (loop count ≥ 2), FDR < 0.01 and loop length ≥ 5000 bp. HiChIP samples passed quality control if ≥4000 significant loops were detected. To identify the differential loops, we defined lost and gained loops in glioblastoma as astrocyte-specific and GB-specific loops, respectively. The criteria were set such as lost loops are those present in normal astrocytes and absent in all GB lines [i.e., ∃ astrocytes (counts ≥ 2) & ∄ 14/14 GB (counts = 0)], while gained loops are absent in normal astrocytes and present in at least 8 out of the 14 GB lines [i.e., ∄ astrocytes (counts = 0) & ∃ 8/14 GB (counts ≥ 2)]. Annotation of the significant and differential loops was done using GenomicInteractions package in R (1.26.0). Loops were categorized as promoter-promoter (P-P), promoter-enhancer (P-E) or enhancer-enhancer (E-E) setting the criteria for promoter regions <2 kb TSS and considering E-E as distal-distal regions. Differences in loop length (bp) between lost and gained loops were assessed using a two-sided $t$-test with Benjamini-Hochberg correction. Among the differential loops, multi-anchor loops were identified as loops that share anchors with other loop(s) i.e., one anchor is utilized by two or more loops. Search for TFBS (transcription factor binding sites) motifs was conducted using HOMER (v4.11) tool findMotifsGenome.pl. Motif analysis was performed within the anchors of the differential lost and gained loops. Significantly enriched TF motifs ($p < 0.01$) were intersected with the 497 DEGs to identify transcription factors that were differentially expressed and whose motifs were enriched at the anchors of the differential loops.

## CUT&RUN

SMAD3 and PITX1 genome-wide binding sites were determined in 13 patient-derived GB lines, normal human astrocytes and human OPCs using the CUT&RUN Assay kit (Cell Signaling Technologies, #86652) and following the manufacturer's instructions. $10^5$ cells per line and per CUT&RUN reaction were collected and mildly fixed in 0.1% formaldehyde for 2 min at room temperature on a shaker, and cross-linking reaction was quenched by adding Glycine (final concentration 125 mM glycine) for 5 min. The cell suspension was first incubated with the concavalin beads, and further incubated with either SMAD3 (ab208182, 4 µg per reaction) or PITX1 (sc-271435, 4 µg per reaction) antibodies for 2 h at 4 °C. Thereafter, chromatin-bound beads were mixed with pAG-MNase in digitonin buffer, and the pAG-MNase enzyme was activated by adding cold calcium chloride and incubated at 4 °C for 30 min. Decrosslinking was performed by incubation with RNase at 37 °C for 10 min followed by proteinase K treatment at 65 °C for 2 h. Enriched DNA was purified using the DNA purification kit (Cell Signalling Technologies #14209 S) and further processed for library preparation using the DNA library Prep Kit for Illumina (Cell Signaling Technologies, #56795). Library size distribution was assessed via Bioanalyzer High Sensitivity DNA Analysis (Agilent #5067-4626) and libraries were sequenced as PE150 on a NovaSeq6000 Sequencing System (Illumina).

**CUT&RUN analysis.** CUT&RUN analysis was performed according to pipelines described before[69]. Fastq files were quality-checked with FastQC (https:// www.bioinformatics.babraham.ac.uk/projects/fastqc/, 0.11.9), paired-end reads were trimmed using TRimmomatic (v 0.39) to remove Illumina adapters and then aligned to the human genome (GRCh38/hg38) using bowtie2 (2.4.5) (--local --very-sensitive-local --no-unal --no-mixed --no-discordant --phred33 -I 10 -X 700). Duplicates were marked using MarkDuplicates from Picard toolbox (2.27.5) and filtered with samtools (1.17). Peak calling was performed using SEACR (1.3) (FDR < 0.01 norm stringent). To identify SMAD3 and PITX1 differential peaks between GB and control lines (astrocytes and OPCs) we

used bedtools multiinter (2.30.0) and segregated the intersected regions in R with dplyr (1.1.3). For both SMAD3 and PITX1, we defined GB-specific as those regions present in GB and absent in both astrocytes and OPCs [i.e., ∃ 6/13 GB & ∄ astrocytes & ∄ OPCs], and control-specific regions as those present in either of the control lines and absent in all GB lines [i.e., ∃ astrocytes/OPCs & ∄ 13/13 GB].

## Mice, surgical procedures and MRI

All experiments with laboratory animals were performed in compliance with national and institutional laws, and according to protocols approved by the Regional Ethics Committee at the Court of Appeal of Northern Norrland (ethical permit ID A29-2019 and A3-2023). Mice were housed under 12:12 h light:dark cycle conditions in temperature- and humidity-controlled rooms (22 °C and 50% humidity). Tumour formation was induced by intracranial injection of $10^5$ patient-derived GB cells (U3013) in neonatal NSG mice (*Mus musculus*, NOD.Cg-*Prkdc*$^{scid}$ *Il2rg*$^{tm1Wjl}$/SzJ, P1-P2). Mice of both sexes were used in this study (given the use of mice at neonatal stages, an early time-point at which sex is not assessed, all pups were orthotopically transplanted to have littermate controls and representation from both sexes). Tumour progression was monitored longitudinally by MRI (Magnetic Resonance Imaging) scanning once a week starting at week 4 after tumour induction. Animals were randomly divided into two groups and were administered either the SMAD3 inhibitor SIS3 (15 mg/kg in 7% DMSO, 30% PEG300, 2% Tween80) or the vehicle by oral gavage at 4.5 and 6.5 weeks after tumour induction. For MR imaging, mice were anesthetized using isoflurane (4% for induction and 2% for anaesthesia maintenance). Animals were kept on a heating pad and respiratory frequency and body temperature were monitored throughout the procedure. The imaging was performed in a Bruker MR scanner (Bruker BioSpec 94/20, running Paravision 7.0 software) using a T2-weighted TurboRARE sequence (TR = 3.6 s; TE = 37 ms; RARE factor = 8; matrix size = 128 × 96; field of view = 12.8 × 9.6 mm; slice thickness = 0.2 mm). All the images were exported as dicom files and volumes were manually calculated using the 3D Slicer software (v 4.11). The volume of the lesions was determined as (a) the tumour volume in the medial part of the brain plus the volume of the lateral pathological areas (which include tumour and can include excess cerebrospinal fluid) and, (b) only the volume of the tumours in the medial part of the brain. Maximal tumour size permitted by the ethics committee was not exceeded (10 mm). Mice were euthanized by carbon dioxide ($CO_2$) inhalation at the endpoint ($t$ = 8 weeks).

## Immunofluorescence

For immunofluorescence, cells were cultured as described above, fixed in 4% formaldehyde for 15 min at room temperature, permeabilized in 0.2% Triton-X100 for 5 min and subjected to antibody incubation. Images were taken using a Nikon eclipse E800 microscope (Fig. 7c) or a Leica widefield Thunder microscope (Supplementary Fig. 10k–o). Antibodies used were as follows: β3-tubulin (Abcam, ab18207, 1:1000), SMAD3 (Abcam, ab208182, 1:500), PITX1 (Santa Cruz, sc-271435, 1:200), TNIK (Abcam, ab224252, 1:1000), EPHB3 (Abcam, ab133742, 1:1000) and KCNE4 (Abcam, ab254642, 2 µg/ml). Fiji (Image J v1) was used to quantify nuclear SMAD3 levels in 15 patient-derived GB lines and normal astrocytes. Five fields per line were imaged and nuclear SMAD3 levels were measured on individual nuclei upon segmentation of the nuclear area based on the DAPI signal.

## Western-Blot

Whole-cell extracts were prepared in lysis buffer containing 2% SDS and 0.1 M Tris-HCl pH 6.8. Subcellular fractionation was performed using the Pierce™ NE-PER® Nuclear and Cytoplasmic Extraction Reagent Kit (ThermoFisher, #78833). Protein concentration was determined using the Pierce™ BCA Protein Assay Kit (Thermo

Scientific) and measuring absorbance at 560 nm with the Biosan HiPo MPP-96 microplate photometer. Equal amounts of protein were run in Mini-PROTEAN TGX gels (Bio-Rad #4568124) followed by western blotting. Primary antibodies used were as follows: SMAD3 (Abcam, ab208182, 1:1000), PITX1 (ThermoFisher, A300-577A-T, 1:1000), TNIK (Abcam, ab224252, 1:1000), EPHB3 (Abcam, ab133742, 1:1000), Histone H3 (Abcam, ab176842) and GAPDH (Cell Signaling, #14C10). Peroxidase AffiniPure secondary antibodies (Jackson ImmunoResearch, 111-035-003, 1:10,000) and ECL substrate were used for signal detection. Target protein expression was normalised to the stain-free total protein measurement using the Image Lab™ Software (Bio-Rad). Uncropped blots are displayed in Supplementary Fig. 14.

### Protein expression in tissue sections
Protein expression data from glioma was retrieved from the Human Protein Atlas (HPA) (https://www.proteinatlas.org/humanproteome/pathology). We obtained protein expression patterns from High-Grade Glioma (HGG) and Low-Grade Glioma (LGG) for SMAD3, PITX1 and some of their downstream targets of relevance in this study, such as TNIK, KCNE4, EPHB3, CNTNAP1, SEMA5A, SLC26A11, RGS17, CDKN2C, NKX2-2, DUSP10, TMX4, SGSH. We represented the protein expression annotated by HPA as high, medium or low expression for each of the above-mentioned proteins and for all the HGG and LGG samples available, together with representative images.

### Quantitative RT-PCR
Total RNA was extracted using the RNeasy Plus Mini Kit (ID: 74134) and retrotranscribed with RevertAid H Minus Reverse Transcriptase (#EP0451, Thermo Fisher) using random hexamers. A CFX Connect Real-Time PCR Detection System (Bio-Rad) was used to determine relative gene expression levels and *HPRT* was used as a reference gene. qPCR primers used are listed in Supplementary Data 5.

### Statistical analysis
Statistical tests used are indicated in the corresponding Methods sections and in figure legends. Differences in loop length (bp) between lost and gained loops were assessed using a two-sided *t*-test with Benjamini-Hochberg correction. Kaplan-Meier survival curves were plotted for patients stratified by quartiles of expression, using TCGA data publicly available at Xena browser (https://xenabrowser.net), and tested using a log-rank Mantel-Cox test. Differences in SMAD3/PITX1 motif distance were assessed using a two-sided Wilcoxon test. Differences in cell proliferation were determined using unpaired two-sided *t*-test with correction for multiple testing ($*p < 0.01$, $**p < 0.005$, $***p < 0.000005$). Differences in tumour volume were assessed using a two-sided Mann Whitney test with correction for multiple testing. Normality tests were conducted prior to selecting the most appropriate statistical test for each analysis.

### Bioinformatic analyses and graphics
Most statistical analysis related to RNA-seq, ChIP-seq, ATAC-seq, HiCHIP and CUT&RUN were performed within the R environment (4.0.0–4.3.0) using basic built-in functions and publicly available packages dplyr, plyr, reshape2, ggmisc, ggpubr (R-CRAN). DESeq2, ComplexHeatmap, ClusterProfiler, BSgenome.Hsapiens.UCSC.hg38, ChIPpeakAnno, ChIPseeker and EnsDb.Hsapiens.v86 are open-source tools available via Bioconductor (https://www.bioconductor.org/, 3.11–3.16). Data in Fig. 1f, g, j; 3c, e, h; 6c; 7e, g, j and supplementary figs. 5a; 10b, d, f, h, j; 11; 12c–f and 13a–d were plotted using GraphPad Prism 9. The Circos Plot in Fig. 4a was generated using RCircos (R-CRAN package: 1.2.0). All other plots were created using basic R graphical interface (4.0.0–4.3.0), and ggplot2 (3.4.3). Genomic snapshots were downloaded from UCSC Genome Browser visualizations (https://genome.ucsc.edu/), for which bedgraphs were generated using samtools (1.12) and bamCoverage tool (deepTools- 3.1.0).

### Reporting summary
Further information on research design is available in the Nature Portfolio Reporting Summary linked to this article.

## Data availability
The raw RNA-seq, ChIP-seq, ATAC-seq, HiChIP and CUT&RUN datasets generated in this study have been deposited in GEO (Gene Expression Omnibus) under the accession number GSE217349 (https://www.ncbi.nlm.nih.gov/geo/query/acc.cgi?acc=GSE217349). All sequencing data generated in this study has been mapped to the GRCh38/hg38 human genome. The publicly available GB and LGG TCGA data was retrieved from The Cancer Genome Atlas (TCGA) data portal (https://portal.gdc.cancer.gov/). The remaining data are available within the Article, Supplementary Information or Source Data file. Source data are provided with this paper.

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

## Acknowledgements

We would like to thank Louella Vasquez and other experts at the National Bioinformatics Infrastructure Sweden (NBIS) at SciLifeLab for bioinformatics advice. The computations were enabled by resources in projects snic2020-15–157, snic2021-22-504 and naiss2023-22-55 provided by the Swedish National Infrastructure for Computing (SNIC) and the National Academic Infrastructure for Supercomputing in Sweden (NAISS) at UPPMAX, funded by the Swedish Research Council through grant agreements no. 2018-05973 and no. 2022-06725, respectively. We also acknowledge the Biochemical Imaging Center at Umeå University and the National Microscopy Infrastructure, NMI (VR-RFI 2019-00217) for providing assistance in microscopy; the Small Animal Research and Imaging Facility (SARIF) at Umeå University for assistance with the MRI equipment and image analysis; Ingela Lundberg for help with drug administration by oral gavage, Helena Edlund for providing the pSPAX and pMD2.G plasmids and the anti-GAPDH antibody; and Francesca Aguiló for the anti-H3 antibody. Research in S.R.'s laboratory is supported by the Knut och Alice Wallenbergs Stiftelse (WCMM, Umeå) (S.R.), the Swedish Research Council (2019-01960) (S.R.), the Swedish Cancer Foundation (21 1720) (S.R.), the Kempe Foundation (SMK-1964.2) (S.R.), as well as the Cancer Research [AMP 19–977 (S.R.); AMP 22–1091 and AMP 23–1137 (I.N., S.R.)] and Lion's Cancer Research Foundations in Northern Sweden (L.P 21-2290) (S.R.).

## Author contributions

C.C. and I.N. designed, performed, analysed and interpreted most of the experiments with contributions from C.V. and A.-C.H. C.V. performed and analysed the live-cell imaging and A.-C.H. performed the in vivo experiments. A.H. co-supervised some bioinformatic analysis, and contributed to data interpretation and manuscript editing. S.R. designed and supervised the study, secured funding, analysed and interpreted the data, and wrote the manuscript with input from the other authors.

## Funding

## Competing interests

The authors declare no competing interests.
