## [Peer Review File · Nature Communications]

Rewiring of the promoter-enhancer interactome and regulatory landscape in glioblastoma orchestrates gene expression underlying neuroglial synaptic communicationReviewers' Comments:

Reviewer #1:

Remarks to the Author:

The manuscript by Chakraborty et al describes an integrated epigenetic/transcriptome characterization of 15 GBM lines. By comparing these lines to a human astrocyte line, they found widespread upregulation of genes associated with synapses, and particularly with glutamate receptor signaling. The authors then go on to study two transcription factors – SMAD3 and PITX1 – whose DNA recognition motifs are enriched in accessible chromatin in GBM cells. They observe that SMAD3 inhibition combined with a chemical that induces neuronal activity results in increased GBM cell proliferation when grown in co-culture with iPSC-derived glutamatergic neurons; on the contrary, the same treatment results in decreased GBM cell proliferation in GBM monoculture.

Overall, the work described in this manuscript is not novel, given the sophisticated papers that have been published on the subject of the neuroscience of GBM over the last 4-5 years. The core of the data generated consists of bulk epigenomic and transcriptomic datasets generated from 15 GBM lines that are commercially available. No work was done with primary tumor samples. The bulk omics techniques used are insufficient to address the intra-sample heterogeneity of GBM, including cell lines. Please see the work by Richards et al (Nature Cancer) for a comprehensive study of dozens of GBM primary cultures and tumor samples with scRNA-seq. The same is true for ATAC-seq; please refer to Guilhamon et al (eLife) for a description of intra-tumoral and intra-sample heterogeneity identified by scATAC-seq. The techniques used to characterize the 15 lines are not adequate based on recent literature and on the known complexity of GBM tumors and models.

The choice of an astrocyte line as control is puzzling. The authors state that astrocytes “are considered to be a cell of origin for glioblastoma” (line 77). The most likely cell of origin for GBM is an oligodendrocyte progenitor cell (OPC), as highlighted in many papers using a plethora of different models. Or maybe another progenitor cell in the glial lineage.

The choice of astrocytes as control then biases the outcomes of the omics studies. It is not surprising that synapses-related genes are differentially expressed between GBM and astrocytes, based on extensive work from the labs of Monje, Deneen and others.

In this respect, no validation of the genes identified in the omic assays has been performed. For instance, lines 137-138 should be supported with IF and IHC studies on the DEGs.

Even the two key transcription factors in this study – SMAD3 and PITX1 – are not validated in any way. The authors should do at least immunofluorescence in GBM lines and IHC in primary specimens for these two proteins. ChIP-seq for both proteins should be performed in at least 2-3 GBM patient-derived lines. The identification of DNA recognition motifs does not mean that SMAD3 and PITX1 actually bind to all those regions. ChIP-seq data will clarify their roles in GBM.

In terms of SMAD3, if its associated motifs are enriched in active chromatin in GBM, why is it that inhibiting SMAD3 results in faster GBM cell proliferation? That seems counterintuitive.

The observations on SMAD3 and PITX1 should be validated with knockdown and overexpression studies in patient-derived models. In vitro and in vivo (orthotopic transplantation) assays should be performed with these KD and OE derivatives. Tumor growth rates and effects on mouse survival need to be determined. The observations with the co-culture method (which are potentially very interesting) need to be validated by administering the compounds to mice bearing xenografts.

The classification of GBM in the 4 molecular subgroups - ie Classical, Mesenchymal, Neural and Proneural - is archaic. The Neural subgroup was the result of the inclusion of samples with very low tumor cell content in the original studies. This was acknowledged early on (see for instance Gill et al

PNAS PMID: 25114226 and Sturm et al 2012) and the Neural subgroup was not considered any more. Mario Suva has proposed a new nomenclature based on single-cell studies based on 4 categories: OPC-like, NPC-like, MES-like and AC-like. Pugh and Dirks proposed a largely similar but more linear nomenclature, with Developmental (corresponding to OPC and NPC-like) and Injury Response (corresponding to MES and AC-like). These newer nomenclatures better reflect the fact that each tumor – often even a GBM culture – include cells with different transcriptional profiles. This co-existence of cell types negates the use of bulk omics methods for epigenetic and transcriptomic classification of GBM models.

Given that the authors had access to 15 patient-derived lines, it is not clear why they used U251 for in vitro assays. This line is considered a poor model of GBM and should not be used.

Fig 1i is supposed to show that GBM and LGG stratify separately, but there is some intermixing of the tumor types. Please perform statistical analyses to address the robustness of the proposed clustering. This is very important, especially because there are so many more LGGs than GBM samples.

Line 126: Differential expression analysis was performed for GBM and LGG “with respect to the corresponding controls.” What are these controls?

Figures 3d-f completely lack statistics. Please statistically assess the differences between GBM and LGG.

Do the survival curves in Fig 5c include both GBM and LGG patients in the TCGA cohort?

MINOR COMMENTS

- Line 23: The WHO 2021 guidelines changed grading from Roman to Arabic numerals. Please change “grade IV” to “grade 4.”
- The acronym GB is defined early on, but then the authors keep referring to “glioblastoma.”
- Line 83: The authors mention “binding sites for histone modifications.” Histone modifications do not bind to anything, but rather they are chemical modifications of histones, and their enrichment along the genome can be assessed by ChIP-seq. Please correct the text.
- Line 147: “Enhancer mark H3K4me3.” This mark is not very good for enhancers. Current standards indicate that H3K4me1 and H3K27ac co-occupancy, plus Mediator subunits and/or BRD4, mark active enhancers.

Reviewer #2:

Remarks to the Author:

The authors have done a comprehensive study of multiple patient-derived glioma cell lines and profiled them using different high throughput genomic techniques. The authors then proceeded with deep analysis of the dataset, aiming to identify pathways and gene programs specific to cancer cells and consequently propose suitable targets for therapy. While the study is thorough and interesting, there are several major issues that need to be addressed:

1. The cell-of-origin of gliomas is being debated a lot in recent years and there are reports pointing to different cells of origin, including cell types other than astrocytes, for example OPCs. Therefore, selecting just astrocytes as the control for differential expression analysis can introduce many biases, especially since the cell lines were grown in different media. The authors should try comparing glioma cell lines to additional normal cell types, that can also be cells of origin for gliomas, in order to exclude that most of the observed changes are features of normal astrocytes.
2. From the figure of comparing the cells to astrocytes, the glioma cell lines seem too homogenous and the changes are probably just characterizing astrocytes, while there should be bigger differences across the subgroups of glioma. It would be better to group cell lines by their subtype and do

differential expression and other analysis based on these groups, given the issues with having proper control. Even if the main point of the study is the comparison with control cells, some form of subgroup analysis will be beneficial.

3. Have the authors measured the total levels of histone marks in cells that are being studied? It seems there is no ChIP normalization, which might be an issue if the total levels of histone modifications are different across the cell lines compared.

4. While primary cell lines are good enough for collecting initial data and comparing different subgroups, to get more robust evidence of changes in gene expression, epigenomics and chromatin organization, a better controlled, "isogenic" system would be better. It is hard to suggest such a system without having the mutational profiles of these cells, although most likely they are quite heterogenous which complicates it further.

5. It is positive to see the assessment of differentially expressed genes in the TCGA dataset of primary tumors. While cell lines are often good models for studying cancers, it is important to replicate the findings in tumor dataset. Building on this, the authors could try to obtain or generate additional datasets for tumor samples, comparing to their epigenomic and chromatin organization results.

6. It would be nice to have a figure panel with mutational profile of the 15 cell lines used, along with subtypes of glioma and additional features as needed. That would help to understand/visualize the dataset (panel of cell lines) better.

Reviewer #3:

Remarks to the Author:

In this manuscript, the authors map the promoter-enhancer interactome and other regulatory landscape of glioblastoma in 15 patient-derived glioblastoma lines that represent various disease subtypes. A very broad and informative multi-omics characterization is performed. Normal brain astrocytes are taken as control, and comparisons are made between glioblastoma subtypes, and to LGG. A loss of long-range regulatory interactions and overall activation of promoter hubs is found. Interestingly, the authors find a specific regulation of synapse-related genes, particularly glutamatergic synapses, with the transcription factors SMAD3 and PITX1 as major direct regulators involved.

All in all, this manuscript is a very interesting addition to the emerging field of brain cancer neuroscience, and provides fascinating guidance about the gene regulatory underpinnings of the neural transcription programs involved in neuroglial synaptic signalling in particular, and general neural features of glioblastoma. The manuscript is very well written, the experiments are sound, and the interpretation of the data is solid, as is the statistics.

The key findings can serve as a roadmap for further development of the field, particularly with respect to the search for master regulators of this particular biology of incurable gliomas.

I would recommend the following additions and modifications to make this study even stronger:

LGGs are used as a comparison (Fig. 1 for example). However, it has been reported before that LGGs also receive neuron-glioma synaptic input: IF they are incurable = diffuse 1p/19q intact astrocytomas. In contrast, no neuron-glioma synapses could be detected yet in oligodendrogliomas. Do the authors find differences between LGGs depending on the 1p/19q status with respect to relevant parameters investigated here?

Likewise, normal brain astrocytes (immortalized?) are used as the key control/comparison for this study. This is understandable; indeed, glioblastoma cells share a lot of cellular/biological similarities with normal astrocytes. However, it would be informative to learn whether key findings also hold true when comparing glioblastoma with e.g. oligodendroglioma (see above), or other cancer entities, particularly curable ones inside or outside the brain (where functional and growth-relevant neuron-

cancer synapses are at least much less likely, if not ruled out so far).

I would recommend to make explicitly or implicitly clear (abstract, introduction, early result section) that this is NOT a hypothesis-driven study specifically looking at "cancer neuroscience" aspects of glioblastoma - but that, to the contrary, the findings reported here are primarily not-hypothesis-driven/unsupervised results, and/but fit very well to the recent exciting biological findings in the field.

The authors should also discuss the manuscript of Osswald et al Nature 2015 and Jung et al J Neurosci 2017 here, and ideally should check whether the neurogenesis/neurodevelopmental pathways of glioblastoma described here, and which are crucial for the formation of these neurite-like extensions of glioblastoma cells that are highly important for receiving neuro-gliomal synaptic input, can also be re-located in their datasets and analyses.

The same is true for the current paper of Hausmann et al. Nature 2022/2023; the authors report loss of repression of genes involved in calcium ion homeostasis, which would go into this direction: any genes here related to KCa3.1 signaling, pacemaking, or intercellular calcium wave propagation?

Manuscript: *Rewiring of the promoter-enhancer interactome and regulatory landscape in glioblastoma orchestrates gene expression underlying neuroglial synaptic communication.*

REVIEWERS' COMMENTS

AUTHORS' POINT-BY-POINT RESPONSE

Reviewer #1, expertise in brain cancer epigenomics and models (Remarks to the Author):

The manuscript by Chakraborty et al describes an integrated epigenetic/transcriptome characterization of 15 GBM lines. By comparing these lines to a human astrocyte line, they found widespread upregulation of genes associated with synapses, and particularly with glutamate receptor signaling. The authors then go on to study two transcription factors – SMAD3 and PITX1 – whose DNA recognition motifs are enriched in accessible chromatin in GBM cells. They observe that SMAD3 inhibition combined with a chemical that induces neuronal activity results in increased GBM cell proliferation when grown in co-culture with iPSC-derived glutamatergic neurons; on the contrary, the same treatment results in decreased GBM cell proliferation in GBM monoculture.

Overall, the work described in this manuscript is not novel, given the sophisticated papers that have been published on the subject of the neuroscience of GBM over the last 4-5 years. The core of the data generated consists of bulk epigenomic and transcriptomic datasets generated from 15 GBM lines that are commercially available. No work was done with primary tumor samples. The bulk omics techniques used are insufficient to address the intra-sample heterogeneity of GBM, including cell lines. Please see the work by Richards et al (Nature Cancer) for a comprehensive study of dozens of GBM primary cultures and tumor samples with scRNA-seq. The same is true for ATAC-seq; please refer to Guilhamon et al (eLife) for a description of intra-tumoral and intra-sample heterogeneity identified by scATAC-seq. The techniques used to characterize the 15 lines are not adequate based on recent literature and on the known complexity of GBM tumors and models.

We acknowledge all the points raised by the reviewer which have helped us improve our manuscript and have reinforced our conclusions with the new data obtained, which includes the addition of another control line (i.e. OPCs: oligodendrocyte progenitor cells, and multi-omics data on OPCs), additional sequencing data (i.e. CUT&RUN to map TFs binding sites in 13GB lines, astrocytes and OPCs) and *in vivo* validations in PDX in mice, among others.

We have now included the abovementioned references (Richards *et al* Nature Cancer and Guilhamon *et al* eLife, refs 24 and 31 respectively, lines 76 and 78), which together with other referenced studies have elucidated the heterogeneity of GB tumors using single-cell omics approaches. We want to emphasize that the aim of our study was not to reproduce or repeat this type of studies, which undoubtedly are very relevant in the field, but instead we aimed to find common mechanistic features across the broad spectrum of GB tumors, since they are heterogeneous but an entity distinguishable from low-grade gliomas. For this purpose, in this unsupervised study, we performed a multi-omics approach in which we characterized not only the transcriptome, chromatin accessibility and the regulatory landscape, but we also used other sophisticated techniques such as HiChIP to map the promoter-enhancer interactome, in a panel of patient-derived GB lines that represent the broad spectrum of GB tumours. In addition, we have now added other state-of-the-art approaches such as CUT&RUN in the revised manuscript.

Single-cell and bulk sequencing approaches address different questions at different resolution and precision, inherent to either method, and our choice was based on our specific biological question. The aim was not to address the heterogeneity at single-cell level, for which purpose single-cell analysis are a better choice. Our focus was to find common features in terms of expression, chromatin accessibility, enhancer landscape and 3D chromatin organization (i.e. promoter-enhancer interactome), for which bulk sequencing approaches are not only valuable but also appropriate to use. Bulk approaches are widely used, reliable and well-established, they have high statistical power and can provide valuable insights into the overall biology of the samples. Given our question, bulk sequencing was even a preferred choice since GB cells transition between different states (Neftel *et al* Cell 2019). The transitions between states and the co-existence of subtypes precisely favors the choice of bulk omics

since, as mentioned, the aim was to identify common mechanistic features in different GB disease states/subtypes.

The novelty of this study lies on the findings: a rewiring of the regulatory and topological landscape that leads to transcriptional changes related to the neuroglial synaptic communication. Moreover, we point to the regulatory networks involved and we functionally validated these findings (both in co-culture systems and *in vivo*). To our knowledge, this study is the first to shed light on the regulatory networks mediating the neuroglial synaptic communication.

The choice of an astrocyte line as control is puzzling. The authors state that astrocytes “are considered to be a cell of origin for glioblastoma” (line 77). The most likely cell of origin for GBM is an oligodendrocyte progenitor cell (OPC), as highlighted in many papers using a plethora of different models. Or maybe another progenitor cell in the glial lineage.

The choice of astrocytes as control then biases the outcomes of the omics studies. It is not surprising that synapses-related genes are differentially expressed between GBM and astrocytes, based on extensive work from the labs of Monje, Deneen and others.

We agree with the reviewer that, even if glioblastoma is thought to originate from glial cells, the particular cell of origin of GB remains under study (Habib *et al* Stem Cell Rev and Rep 2022). The generally accepted view is that there are three cells of origin of GB: neural stem cells (NSCs), NSC-derived astrocytes and oligodendrocyte progenitor cells (OPCs). This is based on cell surface marker expression, cell morphology and gene profiles in GB similar to those normal cell types (reviewed Yao *et al* Cell Mol Imm 2018); together with mouse genetic modeling of GB showing that many CNS cell types are able to develop into GB tumors: such as NSCs (Zhu *et al* Cancer Cell 2015, Lee *et al* Nature 2018), NSCs-derived astrocytes (Chow *et al* Cancer Cell 2011, Friedman *et al* Science 2012) and OPCs (Liu *et al* Cell 2011). Moreover, astrocytes are physically connected to each other through gap junctions and glioma cells are similarly interconnected through tumour microtubes that propagate calcium signals across the glioma network (Osswald *et al* Nature 2015, Hausmann *et al* Nature 2023), further highlighting the similarities between astrocytes and glioma cells. Astrocytes also regulate synapse formation, function and plasticity (Lyon and Allen, Front. Neural Circuits 2022).

Based on this, we think that our choice of normal astrocytes as a control can be considered as adequate. However, to reinforce our original conclusions we have now included in the revised manuscript additional data generated in OPCs, also considered a cell of origin for GB, as suggested by the reviewers. The data generated in OPCs includes RNA-seq, ChIP-seq for H3K27ac and H3K27me3 histone marks, ATAC-seq and CUT&RUN to map SMAD3 and PITX1 binding sites, and it supports the conclusions in the original manuscript:

1. Comparing the transcriptome profiles of the 15 GB *versus* the OPCs, we also detected changes in the expression of genes related to synapse organization, glutamate signaling, axon guidance/axonogenesis and DNA-binding/chromatin remodeling, as we had observed in comparison to astrocytes. We detected expression changes in these genes related to neural-related processes independently of whether we compared to astrocytes or to OPCs. Moreover, we observed changes in the expression of genes described in recent literature for their relevance in GB pathogenesis, such as those involved in the assembly of neural circuits (Krishna *et al* Nature 2023), autonomous rhythmic activity in glioma networks (Hausmann *et al* Nature 2023), and neurodevelopmental pathways crucial for the formation of tumour microtubes of high importance for receiving neuroglial synaptic input (Osswald *et al* Nature 2015, Jung *et al* J Neurosci 2017). The transcriptome data on OPCs is included in New Fig. 1g-l, New Supplementary Figs. 3-4 and New Supplementary Table S2, and referred in the manuscript lines 145-160. Moreover, among the differentially expressed genes (DEGs) in GB vs OPCs, we find SMAD3 and PITX1 downstream targets that we had previously identified in our original study with astrocytes (New Supplementary Fig. 8a, displayed on response to reviewer 2 point 2).

New Fig. 1i,k,l (full figure available as attachment)

New Supplementary Fig. 4b (full figure available as attachment)

- Profiling of histone modifications (H3K27ac, H3K27me3) and chromatin accessibility (ATAC) in OPCs further supports the changes in the regulatory landscape originally described. This data is included in New Supplementary Fig. 6a,b and New Supplementary Table S3 and referred in manuscript lines 225-236.
- We had observed changes in the regulatory landscape (i.e redistribution of histone marks, chromatin accessibility) and 3D chromatin organization (i.e. promoter-enhancer interactome) at the promoters of genes related to synapse organization and axon guidance/axonogenesis, which in turned resulted in dysregulation of those genes, and SMAD3 and PITX1 motifs were enriched at those sites. We have now mapped the SMAD3/PITX1 binding sites by CUT&RUN in 13 GB lines, astrocytes and OPCs, and determined the differential binding of these TFs at some of these gene promoters. This data is included in New Fig. 6b, New Supplementary Fig. 12a,b and New

Supplementary Table S3, and referred in the manuscript lines 321-324. This is further discussed on page 5.

New Fig. 6b (full figure available as attachment)

In this respect, no validation of the genes identified in the omic assays has been performed. For instance, lines 137-138 should be supported with IF and IHC studies on the DEGs. Even the two key transcription factors in this study – SMAD3 and PITX1 – are not validated in any way. The authors should do at least immunofluorescence in GBM lines and IHC in primary specimens for these two proteins. ChIP-seq for both proteins should be performed in at least 2-3 GBM patient-derived lines. The identification of DNA recognition motifs does not mean that SMAD3 and PITX1 actually bind to all those regions. ChIP-seq data will clarify their roles in GBM.

We thank the reviewer for this comment. We have now included in the manuscript immunofluorescence (IF) and western-blots to validate DEGs, including *SMAD3* and *PITX1*, and others of relevance such as *TNIK*, involved in glutamatergic synaptic function (Coba *et al* J Neurosci 2012), *EPHB3*, one of the classical axon guidance molecules (Yu *et al* Nat Neurosci 2001) and *KCNE4*, encoding for a potassium voltage-gated channel. Western-blots are included in New Supplementary Fig. 10a-h (refer to full figure, not displayed below) and IFs in New Supplementary Fig. 10i-m (displayed below) and referred in the manuscript lines 305-310. Altogether, at protein levels we also observe a downregulation of SMAD3, and upregulation of PITX1, TNIK, EPHB3 and KCNE4 in GB cells. Noteworthy are the changes in TNIK and EPHB3 that are almost undetected in astrocytes but highly expressed in GB, and KCNE4 that also presents higher expression levels in GB.

New Supplementary Fig. 10i-m (full figure available as attachment)

Moreover, we took advantage of the immunostainings in primary specimens from High-Grade (HGG) and Low-Grade Glioma (LGG) patients available at the Human Protein Atlas, and we retrieved the expression of SMAD3 and PITX1 together with 12 of their identified downstream targets in the HGG and LGG tissue samples available. This data is included in New Supplementary Fig. 11a (see below the top panels corresponding to SMAD3 and PITX1, for the other proteins refer to the full New Supplementary Fig. 11a), and referred in the manuscript lines 305-310.

New Supplementary Fig. 11a (full figure available as attachment)

As an alternative to ChIP-seq to map TF binding sites, we used CUT&RUN, which performs well with low number of cells and yields low background. We mapped SMAD3 and PITX1 binding sites in 13 GB patient-derived lines, astrocytes and OPCs. This data is included in New Fig. 6b (displayed below), New Supplementary Fig. 12a,b and New Supplementary Table S3, and referred in the manuscript lines 321-324. We detected SMAD3/PITX1 differential binding at the promoters of downstream target genes that we had previously identified, in GB alongside astrocytes and OPCs.

New Fig. 6b (full figure available as attachment)

In terms of SMAD3, if its associated motifs are enriched in active chromatin in GBM, why is it that inhibiting SMAD3 results in faster GBM cell proliferation? That seems counterintuitive. SMADs can act both as transcriptional co-activators and co-repressors via interaction with various transcriptional regulators (Miyazono *et al* Springer 2006 doi.org/10.1007/1-4020-4709-6_14), and in cancer the epigenome also regulates the context-specific effects of TGF- β /SMAD3 (Vidakovic *et al* Cell Reports 2015). This explains why this sounds counterintuitive at first, and we have now clarified it in lines 424-429. We observed examples where SMAD3 binds to gene promoters in astrocytes/OPCs but not in GB, and this is accompanied by redistribution of histone marks, chromatin accessibility and upregulation of the corresponding genes in GB. This would be suggestive of a repressive role of SMAD3 in some cases, while in some others it seems to act as a transcriptional activator (see examples in New Figure 6b).

The observations on SMAD3 and PITX1 should be validated with knockdown and overexpression studies in patient-derived models. In vitro and in vivo (orthotopic transplantation) assays should be

performed with these KD and OE derivatives. Tumor growth rates and effects on mouse survival need to be determined. The observations with the co-culture method (which are potentially very interesting) need to be validated by administering the compounds to mice bearing xenografts.

We thank the reviewer for these helpful suggestions that have improved our manuscript. We have now validated our observations in the co-culture model with new data obtained from *in vivo* longitudinal studies, where we administered the SMAD3 inhibitor SIS3 to mice bearing PDX (patient-derived xenografts). We induced tumours in mice by intracranial injection of the patient-derived GB line U3013, and administered either vehicle or SIS3 to the control and treatment groups, respectively. We followed tumor progression by MRI on a weekly basis between weeks 4 and 8 after tumour induction and measured the volume of the lesions at these 5 time-points. We observed that SMAD3 inhibition exacerbates the disease progression. This data is included in New Figure 7i-k (displayed below), New Supplementary Fig. 13c,d and is referred in manuscript lines 369-372.

New Fig. 7i-k

In addition, we established KD and OE lines for *SMAD3* and *PITX1* in the GB patient-derived line U3013 (New Supplementary Fig. 12c-h) with the purpose of addressing the effect of knockdown and overexpression in the co-culture assay. Unfortunately, the patient-derived GB lines do not grow in the co-culture medium, meaning this experiment was not possible to conduct. This is likely due to the specific culture requirements of these GB lines established at the HGCC, since they are GSCs cultured under stem cell conditions. See below Reviewers Figure 1a which displays the normal proliferation rates of U3013 GB cells under regular culture conditions, in comparison to 1b (purple line) where U3013 lines do not proliferate in co-culture medium in the 72h time-window. Noteworthy, the presence of glutamatergic neurons (green) partly rescues the defective proliferation of the U3013 GB cells (purple) in the co-culture medium.

Reviewers Figure 1: a) Proliferation rates of U3013 patient-derived GB cells in U3013 media, according to regular growing conditions established by the HGCC resource. b) Proliferation rates of U3013 GB cells in co-culture media, either in co-culture with glutamatergic neurons (green line) or alone in monoculture (purple line).

Still, we took advantage of these lines and determined the effect of *SMAD3* or *PITX1* KD and OE in the expression of several target genes identified. *SMAD3* and *PITX1* knockdown and overexpression lines display changes in the expression of a set of genes, which we had identified as downstream targets and are involved in synapses, axon guidance and other neural functions. This data suggests that these

TFs regulate the expression of these genes, and it is included in New Figure 6c (also displayed below) and referred in manuscript lines 324-326.

New Fig. 6c

The classification of GBM is the 4 molecular subgroups - ie Classical, Mesenchymal, Neural and Proneural - is archaic. The Neural subgroup was the result of the inclusion of samples with very low tumor cell content in the original studies. This was acknowledged early on (see for instance Gill et al PNAS PMID: 25114226 and Sturm et al 2012) and the Neural subgroup was not considered any more. Mario Suva has proposed a new nomenclature based on single-cell studies based on 4 categories: OPC-like, NPC-like, MES-like and AC-like. Pugh and Dirks proposed a largely similar but more linear nomenclature, with Developmental (corresponding to OPC and NPC-like) and Injury Response (corresponding to MES and AC-like). These newer nomenclatures better reflect the fact that each tumor – often even a GBM culture – include cells with different transcriptional profiles. This co-existence of cell types negates the use of bulk omics methods for epigenetic and transcriptomic classification of GBM models.

We agree with the reviewer that the classifications proposed by the labs of Suva and Pugh are more recent and based on single-cell approaches. The reason why we refer to the 4 expression subtypes is because the 15 patient-derived GB lines obtained through the Human Glioma Cell Culture (HGCC) resource were already classified according to these criteria, and consisted of 5 mesenchymal, 5 classical, 3 proneural and 2 neural. Our intention was to have a panel that represents the broad heterogeneity of GB independently of the classification, which is the case of these panel of 15 GB lines, with the purpose to identify common mechanistic features in an unsupervised study. Our observations, in terms of rewiring of the enhancer landscape and 3D organization in GB, are made across this panel that represents the heterogeneity of GB, and not pinpointing to any particular subtype. As already mentioned, since our aim was to find common features across the spectrum of GB, and given the transitions between states and the co-existence of subtypes, the use of bulk sequencing was adequate.

Given that the authors had access to 15 patient-derived lines, it is not clear why they used U251 for in vitro assays. This line is considered a poor model of GBM and should not be used.

We consider that performing the co-culture assays with U251 cells was a strength, since we could functionally validate our observations independently on a cell line other than the 15 GB lines used for the multi-omics approaches. We understand the point raised by the reviewer, we are well aware of the fact that U251 cells were established a long time ago, and for this reason we obtained our U251 cells directly from Sigma since they are authenticated by short tandem repeat (STR)-PCR profiling by the manufacturer. To further strength our findings we tried repeating the co-culture assays using some of the patient-derived GB cell lines in the study, unfortunately they do not proliferate in the co-culture medium (see above Reviewers Figure 1), and therefore this experiment was not possible to conduct. Still, the effect of SMAD3 inhibition observed in the co-culture assays was further corroborated *in vivo* in mice, which we consider should suffice to support our conclusions (New Figure 7i-k and New Supplementary Figs 13c,d).

Fig 1i is supposed to show that GBM and LGG stratify separately, but there is some intermixing of the

tumor types. Please perform statistical analyses to address the robustness of the proposed clustering. This is very important, especially because there are so many more LGGs than GBM samples.

This analysis corresponds to a hierarchical clustering, an unsupervised machine-learning strategy that involves finding hidden patterns or groups in the data. We plotted the sample-to-sample distance for the GB/LGG TCGA samples based on Euclidean distance matrix on the log2 normalized expression of our 497 DEGs. This is represented as a heatmap in New Fig. 2d, that includes new annotations, and where we observed a segregation in two clear blocks separating GB and LGG. The robustness of the proposed clustering lies in the underlying machine-learning clustering algorithm. We have confirmed this with several computational experts at the National Bioinformatics Infrastructure Sweden (NBIS) who further corroborated that our analysis is performed correctly. Numerous studies in the literature use the same method, for instance PMID: 36932241, PMID: 35465400, PMID: 27305450 or PMID: 28356166.

Line 126: Differential expression analysis was performed for GBM and LGG “with respect to the corresponding controls.” What are these controls?

The controls mentioned by the reviewer correspond to normal non-tumour samples available at TCGA together with the expression data from LGG and GB primary specimens. This is now clarified in the methods section (lines 539-541): “The differential expression analysis of either GB samples or LGG samples *versus* the respective normal non-tumour control tissues, all retrieved from TCGA, was performed using DESeq2 (FDR<0.01)”.

Figures 3d-f completely lack statistics. Please statistically assess the differences between GBM and LGG.

These panels correspond to density plots representing the normalized H3K27ac, H3K27me3 and RNA-seq signal (New Fig. 4d-f) around the astrocytes’ chromatin states in GB lines and astrocytes. Chromatin states were identified using ChromHMM, a method that uses Hidden Markov Models and random Bernoulli distribution to robustly identify and define complex chromatin patterns (Ernst & Kellis, Nature Methods 2012). Then, the density plot is generated by deepTools using the epigenome/transcriptome signal as input and counting the frequency of read in 100bp windows in ± 10 kb from the center of the chromatin state previously defined with ChromHMM. This approach to visualize the trend in the occupancy of histone marks and DNA-binding proteins is a well-established way of representing this type of data PMID: 30923384, PMID: 31142745, PMID: 36898992, PMID: 29867216, PMID: 27824029. Even though, we further consulted with computational experts at the National Bioinformatics Infrastructure Sweden (NBIS) that corroborated that this analysis was properly performed.

Do the survival curves in Fig 5c include both GBM and LGG patients in the TCGA cohort?
Yes.

MINOR COMMENTS

- Line 23: The WHO 2021 guidelines changed grading from Roman to Arabic numerals. Please change “grade IV” to “grade 4.”

This has been corrected in line 57: “*Glioblastoma (GB) (WHO grade 4 astrocytoma) is the most...*”

- The acronym GB is defined early on, but then the authors keep referring to “glioblastoma.”

We thank the reviewer for the comment. We have reduced the use of the term glioblastoma and favored the acronym GB in the revised manuscript.

- Line 83: The authors mention “binding sites for histone modifications.” Histone modifications do not bind to anything, but rather they are chemical modifications of histones, and their enrichment along the genome can be assessed by ChIP-seq. Please correct the text.

We fully agree with the reviewer and we apologize for the lapsus. In the revised manuscript we use the terms peaks, regions or enrichment when referring to histone modifications, and use binding sites when referring to positions bound by TFs.

- Line 147: “Enhancer mark H3K4me3.” This mark is not very good for enhancers. Current standards indicate that H3K4me1 and H3K27ac co-occupancy, plus Mediator subunits and/or BRD4, mark active enhancers.

We fully agree with the reviewer and apologize for the misleading wording. We wanted to point out that even if H3K4me3 is mostly enriched at gene promoters, it can also be found at enhancers with a much lower observation frequency than the active enhancer mark H3K27ac (Ernst et al Nature 2011, Ernst et al Nature Protocols 2017). To be precise and avoid confusion, we reformulated the sentence in the revised manuscript: lines 120-122 “(...) and H3K4me3, predominantly associated to gene promoters” and lines 200-201 “(...) we performed HiChIP with an antibody against the promoter mark H3K4me3”.

We thank the Reviewer for the thoughtful comments and helpful suggestions that have substantially helped to improve the manuscript.

Reviewer #2, expertise in brain cancer epigenomics (Remarks to the Author):

The authors have done a comprehensive study of multiple patient-derived glioma cell lines and profiled them using different high throughput genomic techniques. The authors then proceeded with deep analysis of the dataset, aiming to identify pathways and gene programs specific to cancer cells and consequently propose suitable targets for therapy. While the study is thorough and interesting, there are several major issues that need to be addressed:

1. The cell-of-origin of gliomas is being debated a lot in recent years and there are reports pointing to different cells of origin, including cell types other than astrocytes, for example OPCs. Therefore, selecting just astrocytes as the control for differential expression analysis can introduce many biases, especially since the cell lines were grown in different media. The authors should try comparing glioma cell lines to additional normal cell types, that can also be cells of origin for gliomas, in order to exclude that most of the observed changes are features of normal astrocytes.

As elaborated above in response to reviewer #1, we agree that the cellular origin of GB is under debate. The view is that not only astrocytes, but also OPCs and NSCs (neural stem cells) can originate GB tumors as evidenced by several studies showing that all these cell types can develop into GB in mouse models (see above page 2 for further details and references). Following the suggestion, we have now included additional data generated in OPCs to have another normal control line and this supports the conclusions in the original manuscript. The data generated in OPCs includes RNA-seq, ChIP-seq for H3K27ac and H3K27me, ATAC and CUT&RUN for SMAD3 and PITX1.

The following results have been included (for more details and to avoid repetition, please refer to response 1 to reviewer 1 in pages 2-3):

1. Comparing the transcriptome profiles of the 15 GB *versus* the OPCs, we also detected changes in the expression of genes related to synapse organization, glutamate signaling, axon guidance/axonogenesis and DNA-binding/chromatin remodeling, as we had observed in comparison to astrocytes. The transcriptome data on OPCs is included in New Fig. 1g-l, New Supplementary Figs. S3-4 and New Supplementary Table S2, and referred in the manuscript lines 144-160. Moreover, among the differentially expressed genes (DEGs) in GB vs OPCs we find SMAD3 and PITX1 downstream targets that we had previously identified in our original study with astrocytes (New Supplementary Fig. 8a, displayed in the response to the next point).
2. Profiling of histone modifications (H3K27ac, H3K27me3) and chromatin accessibility (ATAC) in OPCs further supports the changes in the regulatory landscape originally described. This data is included in New Supplementary Fig. 6a,b and New Supplementary Table S3 and referred in manuscript lines 225-236.
3. SMAD3/PITX1 binding sites by CUT&RUN in 13 GB lines, astrocytes and OPCs shows differential binding of SMAD3/PITX1 at the promoters of genes related with synapse organization, axon guidance/axonogenesis and other neural functions. This data is included in New Fig. 6b, New Supplementary Fig. 12a,b and New Supplementary Table S3, and referred in the manuscript lines 321-324. This is further discussed on page 5.

Last, we would like to point that astrocytes, OPCs and patient-derived GB lines *have to* be grown in defined media containing the specific growth factors and components required to preserve their specific cell identity.

2. From the figure of comparing the cells to astrocytes, the glioma cell lines seem too homogenous

and the changes are probably just characterizing astrocytes, while there should be bigger differences across the subgroups of glioma. It would be better to group cell lines by their subtype and do differential expression and other analysis based on these groups, given the issues with having proper control. Even if the main point of the study is the comparison with control cells, some form of subgroup analysis will be beneficial.

Our understanding is that the reviewer might have interpreted that the GB lines seem “too homogeneous” based on Fig. 1d (displayed below). We will clarify this.

We performed 15 pairwise differential expression analysis where we observed a few thousands (~6000-9000) DEGs in each GB line vs astrocytes (see all MA plots in New Supplementary Fig. 2a). Then we intersected the DEGs from the 15 pairwise comparisons and obtained 497 genes that are differentially expressed versus astrocytes in each and all of the 15 lines. We had originally showed only the Upset for the intersection in 15 out of 15 lines in Fig. 1c (still maintained), but for clarity we have now included full UpSet plots in New Supplementary Fig. 2b,c).

Fig. 1c

New Supplementary Fig. 2b,c

As represented in New Supplementary Fig. 2b,c, there are many transcriptional changes that are common to only some of the GB lines (SuppFig2b) and many other changes that are exclusive to each line (SuppFig2c), and among these changes only 497 genes are differentially expressed in 15 out of 15 GB lines vs astrocytes, which altogether reflects the heterogeneity of this panel of 15 GB lines. Notice that the heatmap in Fig. 1d (displayed above) represents the expression of only the 497 DEGs in astrocytes and the 15 GB lines. Thus, the fact that in this plot the GB lines look so similar among each other and different from the astrocytes is precisely due to the fact that we are plotting only the 497 genes that differ from the astrocytes in all the 15 lines. We hope this helps visualize that the lines are not too homogeneous; noteworthy, these GB lines retain the expression profiles of the original primary tumors as previously characterized (Xie *et al* 2015).

In addition, as suggested by the reviewer, we performed a differential expression analysis based on the four expression subtypes. This analysis shows that the different subtypes present characteristic

expression profiles that differ from the other subtypes, with 400-1000 DEGs in each pairwise comparison (see Reviewer Figure 2), further supporting that the 15 GB lines in this study are not so homogeneous and retain subtype-specific transcriptional profiles. GO analysis of these subtype-based DEGs alongside the DEGs found with respect to our control lines (astrocytes and OPCs) elucidates enrichment for GO terms related to synapse function, axon guidance/axonogenesis and other neural functions, that one would have missed with only the subtype-based differential expression analysis (see Reviewer Figure 3). Therefore, we believe that our study became strengthened by the addition of the OPCs data, and that our experimental setup using astrocytes and OPCs as control cells constitute a good way of approaching this type of studies.

Reviewer Figure 2: MA plots showing genes differentially expressed (DEGs, red dots) in all the possible pairwise comparisons between the four GB subtypes in this study: mesenchymal, classical, neural, proneural.

guidance and axonogenesis. Moreover, 130 genes differentially expressed in GB vs OPCs overlap with the 497 DEGs originally identified when using astrocytes as control. Among these 130 common DEGs, we found genes associated to the abovementioned neural-related processes, further supporting our original observations. We also detected changes in the expression of genes described in recent literature for their relevance in GB: those involved in the assembly of neural circuits (Krishna *et al* Nature 2023), autonomous rhythmic activity in glioma networks (Hausmann *et al* Nature 2023), and neurogenesis/neurodevelopmental pathways crucial for the formation of tumour microtubes of high relevance for receiving neuroglial synaptic input (Osswald *et al* Nature 2015, Jung *et al* J Neurosci 2017). The OPCs' transcriptome data is included in New Fig. 1g-l, New Supplementary Figs. S3-4 and New Supplementary Table S2, and referred in the manuscript lines 144-160 (some of this data is displayed in the response to reviewer 1 on page 3). Moreover, among the differentially expressed genes (DEGs) in GB vs OPCs, we find SMAD3 and PITX1 downstream targets that we had previously identified in our original study with astrocytes (New Supplementary Fig. 8a).

New Supplementary Fig. 8a

3. Have the authors measured the total levels of histone marks in cells that are being studied? It seems there is no ChIP normalization, which might be an issue if the total levels of histone modifications are different across the cell lines compared.

We thank the reviewer the comment. We would like to point that our ChIP-seq data had been normalized, and we can clarify briefly how this analysis was performed following standard ChIP-seq analysis pipelines. Raw fastq reads were first quality checked using fastqc (no adapter contamination and no read duplication was found) and the mapped using bowtie2. Aligned bam files for each sample and their respective inputs were used for peak calling by MACS2. The peak calling algorithm employed by the MACS2 pipeline to normalize aligned reads is one of the most broadly used tools for calling peaks for histone marks and other DNA-binding proteins (PMID: 28092686, PMID: 34131149, PMID: 35858326). A schematic representation of the complete pipeline is displayed below in Reviewers Figure 4. To normalize aligned reads for peak calling, MACS2 scales aligned reads across ChIP-sample & input (control) libraries. It calculates the background (λ_{BG}) events for tags by estimating the number of events occurring in the sample with respect to total events in the genome & local events (λ_{BG}) by estimating maximum number of events at λ_{BG} , λ_{1000bp} , λ_{5000bp} & $\lambda_{10000bp}$. This helps in identifying and treating local enrichment bias and comparing ChIP-seq samples, when they present different quantities of enrichment or different sequencing depths among libraries. Peaks called in this robust manner are further tested for significance and subjected to false discovery rate testing, where it calculates the empirical enrichment of the defined peak in control against the sample, thereby allowing each IP sample to be normalized to its own input (PMID: 18798982, PMID: 22883957). Even if different cell lines might have different levels of histone, by normalizing each ChIP sample to the corresponding input, one decreases the probability of observing enrichment biased due to differences in overall levels in histone marks. We also follow a strict criterion for peak calling setting $FDR \leq 0.05$, to minimize the number of false positives. Moreover, we further consulted with expert bioinformaticians at the National Bioinformatics Infrastructure Sweden (NBIS), who corroborated the adequacy of our analysis pipeline and deemed it appropriate for the purposes of our experiments.

Reviewers Figure 4: Flowchart of the MAC2 pipeline used for ChIP-seq peak calling. (https://hbctraining.github.io/Intro-to-ChIPseq-flipped/lessons/06_peak_calling_mac2.html)

4. While primary cell lines are good enough for collecting initial data and comparing different subgroups, to get more robust evidence of changes in gene expression, epigenomics and chromatin organization, a better controlled, “isogenic” system would be better. It is hard to suggest such a system without having the mutational profiles of these cells, although most likely they are quite heterogeneous which complicates it further.

We agree with the reviewer that these type of experiments with isogenic controls could be interesting but, given the heterogeneity of the GB lines, it would be difficult to find an appropriate system. Nonetheless, we have generated KD and OE lines for SMAD3 and PITX1 in one of the lines (U3013), and detected transcriptional changes in a set of their downstream target genes related to synapse function, axon guidance/axonogenesis and other neural-related functions. This data suggests that SMAD3 and PITX1 regulate the expression of these genes, and it is included in New Figure 6c (displayed below) and referred in manuscript lines 324-326. Besides, using CUT&RUN we detected binding of SMAD3/PITX1 at the promoters of these genes, further supporting their direct role in their transcriptional regulation (see New Fig. 6b and additional details in response to reviewer 1 on page 5).

New Fig. 6c

5. It is positive to see the assessment of differentially expressed genes in the TCGA dataset of primary tumors. While cell lines are often good models for studying cancers, it is important to replicate

the findings in tumor dataset. Building on this, the authors could try to obtain or generate additional datasets for tumor samples, comparing to their epigenomic and chromatin organization results. We thank the reviewer for acknowledging our assessment of the DEGs in the TCGA dataset. We also think that TCGA is a good resource to validate our results since it provides data from primary tumors, with the advantage of the big sample size (hundreds of samples). Noteworthy, we employed GB patient-derived lines that retain the expression profiles of the original primary tumors as previously characterized (Xie *et al* 2015). In our view, having assessed the expression of our DEGs in hundreds of tumor samples from the TCGA minimizes the need to generate data from additional tumour samples, which even if interesting constitutes a full project itself beyond the scope of our study. Nonetheless, we have further explored this dataset and added additional annotations such as 1p/19q status (co-del or non codel), and also highlighted oligodendrogliomas within the LGG group. Interestingly, GB samples do not preferentially cluster with 1p/19q codel LGGs and oligodendrogliomas, that correspond to the LGGs with longer overall survival and where no neuroglial synapses have been detected so far. This data is now included in New Fig. 2d (displayed below) and referred in manuscript lines 171-178. Furthermore, the expression of SMAD3 and PITX1 downstream targets suffices to accurately segregate GB from LGG. This is included in New Fig. 5f (displayed in response to reviewer 3 point 2) and it is referred in manuscript lines 294-296.

New Fig. 2d

6. It would be nice to have a figure panel with mutational profile of the 15 cell lines used, along with subtypes of glioma and additional features as needed. That would help to understand/visualize the dataset (panel of cell lines) better.

We thank the reviewer for the suggestion. We have now included a scheme that helps visualize the information available via the HGCC regarding the mutational profiles of these lines. This is included in New Supplementary Fig. 1 (displayed below) and referred in manuscript line 112.

New Supplementary Fig. 1

We thank the Reviewer for the comments that have helped improve our manuscript and for acknowledging some of the strengths of our study.

Reviewer #3, expertise in neurogliomal synaptic communication (Remarks to the Author):

In this manuscript, the authors map the promoter-enhancer interactome and other regulatory landscape of glioblastoma in 15 patient-derived glioblastoma lines that represent various disease subtypes. A very broad and informative multi-omics characterization is performed. Normal brain astrocytes are taken as control, and comparisons are made between glioblastoma subtypes, and to LGG. A loss of long-range regulatory interactions and overall activation of promoter hubs is found. Interestingly, the authors find a specific regulation of synapse-related genes, particularly glutamatergic synapses, with the transcription factors SMAD3 and PITX1 as major direct regulators involved.

All in all, this manuscript is a very interesting addition to the emerging field of brain cancer neuroscience, and provides fascinating guidance about the gene regulatory underpinnings of the neural transcription programs involved in neurogliomal synaptic signalling in particular, and general neural features of glioblastoma. The manuscript is very well written, the experiments are sound, and the interpretation of the data is solid, as is the statistics.

The key findings can serve as a roadmap for further development of the field, particularly with respect to the search for master regulators of this particular biology of incurable gliomas.

We truly thank the reviewer for appreciating the novelty of our findings and grasping the interest that they constitute in this emerging field. We greatly appreciate the suggestions that have helped us strengthen the manuscript.

I would recommend the following additions and modifications to make this study even stronger:

LGGs are used as a comparison (Fig. 1 for example). However, it has been reported before that LGGs also receive neuron-glioma synaptic input: IF they are incurable = diffuse 1p/19q intact astrocytomas. In contrast, no neuron-glioma synapses could be detected yet in oligodendrogliomas. Do the authors find differences between LGGs depending on the 1p/19q status with respect to relevant parameters investigated here?

This is a very relevant and interesting point raised by the reviewer. We have further explored this data and added additional annotations such as 1p/19q status (co-del or non codel), and also highlighted oligodendrogliomas within the LGG group. Based on the expression of our 497 DEGs, this unsupervised clustering can accurately segregate GB from LGG in two clear blocks. It is true that a small fraction of LGG tumours cluster in the GB block, however all of those are non-codel LGG (i.e. 1p/19q intact), meaning aggressive and incurable. Also, nearly all oligodendrogliomas cluster in the LGG block. This is interesting in view that no neurogliomal synapses have been detected so far in oligodendrogliomas, as pointed by the reviewer. This data is now included in New Fig. 2d (displayed below) and referred in manuscript lines 171-178.

New Fig. 2d

Following the same unsupervised clustering approach, we also observed that the sole expression of the SMAD3 and PITX1 downstream targets suffices to accurately segregate GB from LGG, which further strengthens our findings. This is included in New Fig. 5f (see below) and it is referred in manuscript lines 294-296.

New Fig. 5f

Likewise, normal brain astrocytes (immortalized?) are used as the key control/comparison for this study. This is understandable; indeed, glioblastoma cells share a lot of cellular/biological similarities with normal astrocytes. However, it would be informative to learn whether key findings also hold true when comparing glioblastoma with e.g. oligodendroglioma (see above), or other cancer entities, particularly curable ones inside or outside the brain (where functional and growth-relevant neuron-cancer synapses are at least much less likely, if not ruled out so far).

We appreciate that the reviewer finds that normal astrocytes are an adequate control, and we agree that comparison with other controls is relevant and informative. As a side note, the astrocytes used are not immortalized, they are primary normal astrocytes.

We have now added data obtained from a new control line, OPCs (oligodendrocyte progenitor cells), in our revised manuscript, which further corroborates our original conclusions. The new data includes RNA-seq, ATAC-seq, ChIP-seq for H3K27ac and H3K27me3, as well as CUT&RUN for SMAD3 and PITX1.

The following results have been included (for more details and to avoid repetition, please refer to response 1 to reviewer 1 in pages 2-3):

1. Comparing the transcriptome profiles of the 15 GB *versus* the OPCs, we also detected changes in the expression of genes related to synapse organization, glutamate signaling, axon guidance/axonogenesis and DNA-binding/chromatin remodeling, as we had observed in comparison to astrocytes. The transcriptome data on OPCs is included in New Fig. 1g-l, New Supplementary Figs. S3-4 and New Supplementary Table S2, and referred in the manuscript lines 145-160. Moreover, among the differentially expressed genes (DEGs) in GB vs OPCs we find SMAD3 and PITX1 downstream targets that we had previously identified in our original study with astrocytes (New Supplementary Fig. 8a, displayed in response to reviewer 2 point 2 on page 14).
2. Profiling of histone modifications (H3K27ac, H3K27me3) and chromatin accessibility (ATAC) in OPCs further supports the changes in the regulatory landscape originally described. This data is included in New Supplementary Fig. 6a,b and New Supplementary Table S3 and referred in manuscript lines 225-236.
3. SMAD3/PITX1 binding sites by CUT&RUN in 13 GB lines, astrocytes and OPCs shows differential binding of SMAD3/PITX1 at the promoters of genes related with synapse organization, axon guidance/axonogenesis and other neural functions. This data is included in New Fig. 6b, New Supplementary Fig. 12a,b and New Supplementary Table S3, and referred in the manuscript lines 321-324. (Further discussed on page 5).

Moreover, when assessing the expression of our 497 DEGs in the TCGA dataset containing LGG and GB primary tumours, we observed that nearly all oligodendrogliomas cluster in the LGG block. This is interesting in view that no neuroglial synapses have been detected so far in oligodendrogliomas, meaning that our dataset can accurately segregate samples with, in principle, fundamental differences in this process. This data is now included in New Fig. 2d (displayed in the previous answer to this reviewer) and referred in manuscript lines 171-178.

I would recommend to make explicitly or implicitly clear (abstract, introduction, early result section) that this is NOT a hypothesis-driven study specifically looking at "cancer neuroscience" aspects of glioblastoma - but that, to the contrary, the findings reported here are primarily not-hypothesis-driven/unsupervised results, and/but fit very well to the recent exciting biological findings in the field. This is a very relevant aspect of our study, which likely was not very clear in the original manuscript, and we thank the reviewer for the recommendation. We took advantage of the reviewer's comment to highlight that this is a non-hypothesis-driven study where our results fit well with the recent findings in the cancer neuroscience field. This is now included in manuscript lines 82 and 97-98.

The authors should also discuss the manuscript of Osswald et al Nature 2015 and Jung et al J Neurosci 2017 here, and ideally should check whether the neurogenesis/neurodevelopmental pathways of glioblastoma described here, and which are crucial for the formation of these neurite-like extensions of glioblastoma cells that are highly important for receiving neuro-gliomal synaptic input, can also be re-located in their datasets and analyses.
Addressed together with the next point (see below).

The same is true for the current paper of Hausmann et al. Nature 2022/2023; the authors report loss of repression of genes involved in calcium ion homeostasis, which would go into this direction: any genes here related to KCa3.1 signaling, pacemaking, or intercellular calcium wave propagation?

We thank the reviewer for raising this point that has allowed us to include additional interesting findings from our data. We retrieved a list of genes relevant for GB pathogenesis from the suggested studies (Hausmann *et al* Nature 2023, Osswald *et al* Nature 2015, Jung *et al* J Neurosci) and we additionally added the very recent work by Krishna *et al* Nature 2023. We extracted the expression of those genes in the 15 patient-derived GB lines alongside astrocytes and OPCs from our RNA-seq dataset and we observed significant gene expression changes. Among those are *THBS1*, encoding thrombospondin-1, involved in the assembly of neural circuits (Krishna *et al*); *KCNN4*, encoding the potassium channel KCa3.1, relevant for autonomous rhythmic activity in glioma networks (Hausmann *et al*); and several other genes crucial for the formation of tumour microtubes of high importance for receiving neuroglial synaptic input (Jung.Winkler.2017g, Osswald.Winkler.2015), including *GAP43* that has also been recently reported to mediate mitochondria transfer from astrocytes to GB cells (Watson *et al* Nature

Cancer 2023). This data is now included in New Fig. 11 (see below) and referred in manuscript 155-160.

New Fig. 11

We thank the reviewer for appreciating the novelty of our findings and for considering that they can open the field with respect to the search for master regulators in this context.

Reviewers' Comments:

Reviewer #1:

Remarks to the Author:

- In the rebuttal, the authors state that "The novelty of this study lies on the findings: a rewiring of the regulatory and topological landscape that leads to transcriptional changes related to the neuroglial synaptic communication. Moreover, we point to the regulatory networks involved and we functionally validated these findings (both in co-culture systems and in vivo). To our knowledge, this study is the first to shed light on the regulatory networks mediating the neuroglial synaptic communication." There is a large body of literature about neuroglial synaptic communication, with key papers published from the Monje, Winkler and Deneen labs (among others) that have reported on this. This angle is therefore not novel.
- Regarding their rebuttal of my criticism on the use of bulk genomic methods, I also strongly disagree that bulk methods are suitable to identify common profiles shared by cells in a sample. In fact, the averaging in bulk methods could result in spurious patterns that are not representative of the biological entities in a sample. That's the whole point of using single-cell technologies.
- Regarding the new supplemental figure 10i-m: The authors should note that IF per se is not quantitative. In addition, the nuclei of the GBM cells are clearly much smaller than the nuclei of the astrocyte cell line; I therefore fail to see a difference in the levels of nuclear SMAD3. There might be more SMAD3 in the cytoplasm of astrocytes, but because of the different cell sizes, it's difficult to properly evaluate these qualitative IF images. Along these lines, the authors should look at pSMAD3.
- Regarding the molecular subtypes, I still think the authors need to use the new nomenclature. This is the standard in the field now. The authors should use bulk RNA or ATAC to derive the cell states based on new nomenclature (Suca or other).
- Regarding U251 and other non-patient-derived cell lines: The standards in the field require the use of patient-derived models. I have a hard time trusting data generated with poor models like U251. I appreciate the results of the xenografts, but more data need to be generated with patient-derived models. As the manuscript is now, it does not meet the basic standards in the neuro-oncology field.
- GBM and LGG samples should be considered separately in the survival curves in Fig 6 (Fig 5 in first submission). They are different diseases with vastly different biology and prognosis.
- Line 112: Please remove all mentions of the 'neural' subtype. As I wrote in my previous comments, this subtype was spurious and is not considered any more even among the TCGA subtypes. Please do consult the literature that addresses this issue.
- The role of SMAD3 in regulating expression of genes associated with synaptic communication is still unclear. The SMAD3 motif was found to be enriched at active regions. However, inhibition of SMAD3 increases the proliferation of GBM cells, especially when neural activity is stimulated. This is very confusing because one would expect that SMAD3 is important for expression of synapsis genes (since it's at accessible/active chromatin), and therefore inhibiting it would be bad for GBM cells. What happens to synapses when SMAD3 is inhibited? Can the data generated with SIS3 be replicated with SMAD3 knockdown or degron-mediated depletion? If SMAD3 is involved in repression of gene expression, this needs to be shown.
- Standards in the field require experiments to be generated with more than one line. Please make sure this standard is applied to all the in vitro data generated for this manuscript.

Reviewer #2:

Remarks to the Author:

The authors have significantly improved the manuscripts and performed additional experiments and analyses which have enhanced their findings. I am generally satisfied with authors' responses, except for point 3. While the authors have provided details of their ChIP-seq data analysis, my concern was about varying levels of histone marks, often seen in gliomas, and this cannot be adequately normalized unless the experiments are done with exogenous chromatin spike-in (from different species) or measuring histone mark levels by proteomic methods. There have been several methods

proposed for normalizing for different levels of histone marks without spike-in, but those methods are far from perfect. Considering the presence of multiple datasets in the paper supporting authors' conclusions, this is not a significant obstacle, but for the follow up studies it would be a useful addition to experimental setup.

Reviewer #3:

Remarks to the Author:

The authors have addressed all my points and questions very well. I have no remaining remarks.

... but one: the authors might want to consider to change the title of Fig. 7 to make clear that mouse / in vivo data are also included here.

Manuscript: Chakraborty *et al.* “Rewiring of the promoter-enhancer interactome and regulatory landscape in glioblastoma orchestrates gene expression underlying neuroglial synaptic communication”.

REVIEWERS' COMMENTS

AUTHORS' POINT-BY-POINT RESPONSE

Reviewer #1 (Remarks to the Author):

- In the rebuttal, the authors state that “The novelty of this study lies on the findings: a rewiring of the regulatory and topological landscape that leads to transcriptional changes related to the neuroglial synaptic communication. Moreover, we point to the regulatory networks involved and we functionally validated these findings (both in co-culture systems and in vivo). To our knowledge, this study is the first to shed light on the regulatory networks mediating the neuroglial synaptic communication.” There is a large body of literature about neuroglial synaptic communication, with key papers published from the Monje, Winkler and Deneen labs (among others) that have reported on this. This angle is therefore not novel.

The breakthrough discoveries of Monje, Winkler and Deneen labs, among others, are beyond any doubt. Still, we believe that our study offers mechanistic insights into the changes in the enhancer landscape and 3D chromatin organization that underlies such synaptic communication, which have not been addressed up to date.

- Regarding their rebuttal of my criticism on the use of bulk genomic methods, I also strongly disagree that bulk methods are suitable to identify common profiles shared by cells in a sample. In fact, the averaging in bulk methods could result in spurious patterns that are not representative of the biological entities in a sample. That's the whole point of using single-cell technologies.

We acknowledge that single-cell technologies are very useful and the method of choice to answer particular biological questions. We have introduced a comment in the discussion mentioning that single-cell approaches can in the future provide additional information in the study of gene regulatory networks in this context (lines 406-407).

- Regarding the new supplemental figure 10i-m: The authors should note that IF per se is not quantitative. In addition, the nuclei of the GBM cells are clearly much smaller than the nuclei of the astrocyte cell line; I therefore fail to see a difference in the levels of nuclear SMAD3. There might be more SMAD3 in the cytoplasm of astrocytes, but because of the different cell sizes, it's difficult to properly evaluate these qualitative IF images. Along these lines, the authors should look at pSMAD3.

Given the known difficulties in measuring pSMAD3 levels in the absence of TGF β stimulation, we opted for assessing nuclear SMAD3 using orthogonal methods. First, we performed a subcellular fractionation and measured SMAD3 in both cytoplasmic and nuclear extracts from normal astrocytes and three GB patient-derived lines. We observed that nuclear SMAD3 levels are lower in GB cells than in normal astrocytes. This data is now included in New Supplementary Figure 10i (also displayed below) and referred to in the revised manuscript in lines 304-307.

New Supplementary Figure 10i

Second, using SMAD3 IF images obtained from 15 GB lines and normal astrocytes, we quantified nuclear SMAD3 levels on individual nuclei upon segmentation of the nuclear area based on the DAPI signal. In agreement with previous observations, SMAD3 nuclear levels are lower in GB than in normal astrocytes. This data is included in New Supplementary Figure 10j (also displayed below) and referred in lines 304-307 of the revised manuscript.

New Supplementary Figure 10j

- Regarding the molecular subtypes, I still think the authors need to use the new nomenclature. This is the standard in the field now. The authors should use bulk RNA or ATAC to derive the cell states based on new nomenclature (Suca or other).

Being aware of the recent nomenclature proposed by Suvà lab, we attempted to assign our 15 GB lines to the cellular states identified by Neftel *et al* 2019. Comparing bulk RNA-seq and scRNA-seq data presents several known challenges due to the intrinsic differences in the technologies that generate biases. We generated TPM counts for our 15 GB lines for comparison with the single-cell dataset (Neftel *et al* 2019, GSM3828672). We transformed the single-cell data as pseudo-bulk as described by Barrett *et al.* 2022, and combined it with our 15 GB lines TPM matrix to perform Principal Component analysis (PCA) on either whole transcriptome or signature genes described in Neftel *et al.*, 2019. As observed in *Reviewers' Fig. 1* (displayed below), the pseudo-bulk data and bulk RNA-seq data segregate from one another due to intrinsic differences in both technical approaches. The PC1-PC2 plots reflect the variation arising from the differences in technical approach, segregating the bulk from the scRNA-seq data, both when the analysis is performed on the whole transcriptome (left) and on the signature genes described (right). This is a known problem in the field, and to the best of our knowledge, there are not yet commonly used and firmly established methods to bypass this bias. Thus, given the problem of deconvolution, inferring cell-type expression profiles from bulk samples still remains challenging. Based on this, we prefer not to assign our GB lines to cellular states and therefore choose to keep the nomenclature defined by the HGCC resource at the time of establishing the patient-derived lines.

Reviewers' Fig. 1: PCA plot displaying PC1 vs PC2 for the whole transcriptome (left) or for the signature genes (right).

- Regarding U251 and other non-patient-derived cell lines: The standards in the field require the use of patient-derived models. I have a hard time trusting data generated with poor models like U251. I appreciate the results of the xenografts, but more data need to be generated with patient-derived models. As the manuscript is now, it does not meet the basic standards in the neuro-oncology field.

This study includes a) multiple *omics* data obtained in a panel of 15 patient-derived GB lines, b) *in vivo* validations in PDX mouse models performed in one of the patient-derived GB lines, c) experiments on *SMAD3* and *PITX1* overexpression and knockdown lines established in one of the patient-derived GB lines, and d) co-culture experiments of glutamatergic neurons and U251-GFP cells.

We only used the U251-GFP line in the co-culture experiments (d) to functionally validate our observations on a cell line independently of the 15 GB lines used in the omics approaches (a). As already addressed, the patient-derived GB cell lines in the study do not proliferate in the co-culture medium, and therefore co-culture with these cells is not possible to conduct. Moreover, our *in vitro* findings obtained with U251 cells were validated *in vivo* in PDX mouse models using a patient-derived GB line (b). Nonetheless, we are aware of the related limitations and have included a statement about this in the revised manuscript in lines 469-472.

- GBM and LGG samples should be considered separately in the survival curves in Fig 6 (Fig 5 in first submission). They are different diseases with vastly different biology and prognosis.

In this figure, the purpose is to compare the survival with the inversely correlated expression of *SMAD3* and *PITX1* in TCGA data displayed in Fig. 6e.

- Line 112: Please remove all mentions of the 'neural' subtype. As I wrote in my previous comments, this subtype was spurious and is not considered any more even among the TCGA subtypes. Please do consult the literature that addresses this issue.

We are aware of this issue. We insist that our observations are drawn from a broad panel that represents the heterogeneity of GB, and with no intention to point to any particular subtype. As explained above, we prefer to maintain the nomenclature assigned by the HGCC resource at the time of establishing the cell lines, given the challenges derived from inferring cell-type expression profiles from bulk samples.

- The role of *SMAD3* in regulating expression of genes associated with synaptic communication is still unclear. The *SMAD3* motif was found to be enriched at active regions. However, inhibition of *SMAD3* increases the proliferation of GBM cells, especially when neural activity is stimulated. This is very confusing because one would expect that *SMAD3* is important for expression of synapse genes (since it's at accessible/active chromatin), and therefore inhibiting it would be bad for GBM cells. What happens to synapses when *SMAD3* is inhibited? Can the data generated with SIS3 be replicated with *SMAD3* knockdown or degron-mediated depletion? If *SMAD3* is involved in repression of gene expression, this needs to be shown.

The role of *SMAD3* in terms of transcriptional activation and repression had been clarified. *SMADs* can act both as transcriptional co-activators and co-repressors via interaction with various transcriptional regulators (Miyazono *et al* Springer 2006 doi.org/10.1007/1-4020-4709-6_14). In cancer, the epigenome also regulates the context-specific effects of TGF- β /*SMAD3* (Vidakovic *et al* Cell Reports 2015). We observed examples where *SMAD3* binds to gene promoters in astrocytes/OPCs but not in GB, and this is accompanied by redistribution of histone marks, chromatin accessibility and upregulation of the corresponding genes in GB, which would be suggestive of a repressive role of *SMAD3* at least in some cases. In some others it seems to act as a transcriptional activator. Examples are shown in New Figure 6b. We agree that investigating the effect on *SMAD3* inhibition on synapse structure and function might be interesting, but we consider that this goes beyond the scope of this study.

- Standards in the field require experiments to be generated with more than one line. Please make sure this standard is applied to all the *in vitro* data generated for this manuscript.

This study has employed 15 patient-derived GB lines for the various *omics* approaches alongside 2 normal control cell types (astrocytes and OPCs). The findings were validated *in vitro* in co-culture assays using U251 cells, and in *SMAD3* and *PITX1* overexpression and knockdown experiments in one of the patient-derived GB lines. Besides, *in vivo* validations in mice were performed in one of the patient-derived GB lines.

We thank the reviewer for all the points raised during the revision that have helped strengthen our manuscript.

Reviewer #2 (Remarks to the Author):

The authors have significantly improved the manuscripts and performed additional experiments and analyses which have enhanced their findings. I am generally satisfied with authors' responses, except for point 3. While the authors have provided details of their ChIP-seq data analysis, my concern was about varying levels of histone marks, often seen in gliomas, and this cannot be adequately normalized unless the experiments are done with exogenous chromatin spike-in (from different species) or measuring histone mark levels by proteomic methods. There have been several methods proposed for normalizing for different levels of histone marks without spike-in, but those methods are far from perfect. Considering the presence of multiple datasets in the paper supporting authors' conclusions, this is not a significant obstacle, but for the follow up studies it would be a useful addition to experimental setup.

We are grateful to the reviewer for all the suggestions that have helped improve our manuscript. Regarding point 3, we understand now that the reviewer was referring to ChIP normalization methods where chromatin from a different species is spiked in. It is true that, even if chromatin spike-in was not traditionally used in ChIP-seq experiments, its use is becoming more and more generalized. In our case, it is important to note that we did not perform a differential enrichment analysis, where one would compare higher or lower enrichment levels at a particular genomic region. Instead, we compared enrichment vs no-enrichment i.e., peaks that were present in one condition and absent in another. Therefore, we consider that this type of comparison is likely less sensitive to differences in levels. We agree with the reviewer that this should not be an obstacle given the various datasets and experiments that support our conclusions. Nonetheless, we agree that chromatin spike-in is an important addition to the ChIP-seq workflow to be considered as a routine in the future.

Reviewer #3 (Remarks to the Author):

The authors have addressed all my points and questions very well. I have no remaining remarks.

... but one: the authors might want to consider to change the title of Fig. 7 to make clear that mouse / in vivo data are also included here.

We thank once again the reviewer for appreciating our findings and for all the insightful comments in the revision process.

We have introduced the suggested change in the title of Figure 7 so that the new title is "SMAD3 inhibition and neuronal activity cooperate to promote proliferation of GB cells in co-culture with glutamatergic neurons and *in vivo* in mice".